# Seafloor evidence for pre-shield volcanism above the Tristan da Cunha mantle plume

Wolfram H. Geissler [1✉], Paul Wintersteller [2,3], Marcia Maia[4], Tonke Strack[3], Janina Kammann[5], Graeme Eagles [1], Marion Jegen[6], Antje Schloemer[1,7] & Wilfried Jokat [1,2]

Tristan da Cunha is assumed to be the youngest subaerial expression of the Walvis Ridge hot spot. Based on new hydroacoustic data, we propose that the most recent hot spot volcanic activity occurs west of the island. We surveyed relatively young intraplate volcanic fields and scattered, probably monogenetic, submarine volcanoes with multibeam echosounders and sub-bottom profilers. Structural and zonal GIS analysis of bathymetric and backscatter results, based on habitat mapping algorithms to discriminate seafloor features, revealed numerous previously-unknown volcanic structures. South of Tristan da Cunha, we discovered two large seamounts. One of them, Isolde Seamount, is most likely the source of a 2004 submarine eruption known from a pumice stranding event and seismological analysis. An oceanic core complex, identified at the intersection of the Tristan da Cunha Transform and Fracture Zone System with the Mid-Atlantic Ridge, might indicate reduced magma supply and, therefore, weak plume-ridge interaction at present times.

[1] Alfred Wegener Institute, Helmholtz Centre for Polar and Marine Research, Am Alten Hafen 26, 27568 Bremerhaven, Germany. [2] Faculty of Geosciences, University of Bremen, Klagenfurter Str. 4, 28359 Bremen, Germany. [3] MARUM—Center of Marine Environmental Sciences, University of Bremen, Leobener Str. 8, 28359 Bremen, Germany. [4] CNRS-UBO Laboratoire Domaines Océaniques, Institut Universitaire Européen de la Mer, 29280 Plouzané, France. [5] Department of Geoscience and Natural Resource Management, København Universitet, Øster Voldgade, 101350 København K, Denmark. [6] GEOMAR, Helmholtz Centre of Ocean Research Kiel, Wischhofstr. 1–3, 24148 Kiel, Germany. [7] Department of Earth and Environmental Sciences, Ludwig-Maximilians-Universität, Theresienstr. 41, 80333 München, Germany. ✉email: Wolfram.Geissler@awi.de

Many intraplate volcanic ocean islands are seen as manifestations of hotspots that indicate the presence of underlying mantle plumes. These islands can be the emergent parts of well-developed shield, postshield, or rejuvenated volcanic edifices[1]. Little is known about the embryonic stages of ocean-island growth, which begin with the very first eruptions of volcanic rocks onto pristine oceanic crust. Only a handful of studies have previously investigated the early stages in the development of intraplate volcanic edifices. All were carried out at various hotspots in the Pacific Ocean. At the Easter Island hotspot, Hagen et al.[2] and Fretzdorff et al.[3] discovered and sampled young volcanic fields. Devey et al.[4] reported on young volcanic cones at the Pitcairn hotspot. Loihi Seamount is another example of a young volcanic edifice situated over the Hawai'i mantle plume[5]. We report here the discovery of relatively young volcanic cones and lava fields west of Tristan da Cunha (TdC) in the less-studied South Atlantic. TdC and the associated Walvis Ridge are key hotspot expressions whose source motion and melting history can contribute to the understanding of fundamental processes of mantle plume dynamics and planetary evolution.

TdC is an intraplate volcanic island in the southern Atlantic Ocean, located at 37°06′S and 12°17′W (Fig. 1), about 450 km east of the Mid-Atlantic Ridge (MAR) and 60 km north of the TdC transform fault and fracture zone (TTFZ) system. The subaerial portion of the island covers an area of ~98 km². It rises 2060 m above sea level to its highest point at Queen Mary's Peak.

Known volcanic activity at TdC dates back to ~1.3 Ma[6]. The most recent subaerial volcanic activity occurred in 1961–1962 close to the island's main settlement, Edinburgh of the Seven Seas[7]. In 2004, earthquake activity and pumice stranding on the island's eastern beaches were attributed to a submarine eruption south of the main island[8–10].

Volcanic rocks on the island are dominated by basanites (trachybasalts in the terminology of Baker et al.[7]), followed by moderate quantities of phonotephrites and minor occurrences of trachyandesite and phonolite[7,11–13]. According to Hicks et al.[14], the main edifice is a complex volcano in its postshield phase consisting of interlayered lava and pyroclastic flows with numerous parasitic volcanic centers on its flanks and coastal plains. There is currently no consensus on the classification of the island volcanoes. While TdC is classified as a shield volcano[15] or composite volcano[7], Nightingale Island is listed as a stratovolcano[15]. According to the scheme of average characteristics of primary individual landforms[16], they can be classified as either oceanic shield volcanoes or composite volcanoes.

TdC and the neighboring Inaccessible and Nightingale Islands, islets and rocks are expressions of a classical hotspot. This hotspot marks the present location of a mantle plume, whose previous activity is associated with the formation of the Walvis Ridge that started with the eruption of the Etendeka–Parana flood basalt province in southwest Africa and eastern South America (Fig. 1) at ~132 Ma[17–20]. The volcanic islands are located on ~22–20 Myr-old oceanic crust of the African Plate[21,22].

Debate continues about whether the mantle plume beneath TdC is an expression of convection of the whole mantle[23–25] or of shallower plate-driven convection[26]. Recent work points to complex interactions between a deep plume and shallow asthenospheric and lithospheric processes[27,28].

The youngest part of the Walvis Ridge (Fig. 1) around the volcanically active islands of TdC was the focus of a major scientific project, ISOLDE (2010–2018). ISOLDE's main objective was to characterize the upper mantle in the search for evidence of a mantle plume beneath the volcanic archipelago by means of geophysical and petrological studies[13,21,29–33]. The geophysical program saw ocean-bottom seismometers and magnetotelluric instruments deployed over a period of 11 months between 2012 and 2013.

In this study, we concentrate on relatively young volcanic and tectonic structures observed on the seafloor around the islands in hydroacoustic data, from multibeam echosounder (MBES) and sub-bottom profiler equipment, acquired during the deployment and recovery of ocean-bottom equipment (see "Methods" section and Supplementary information for further information). These

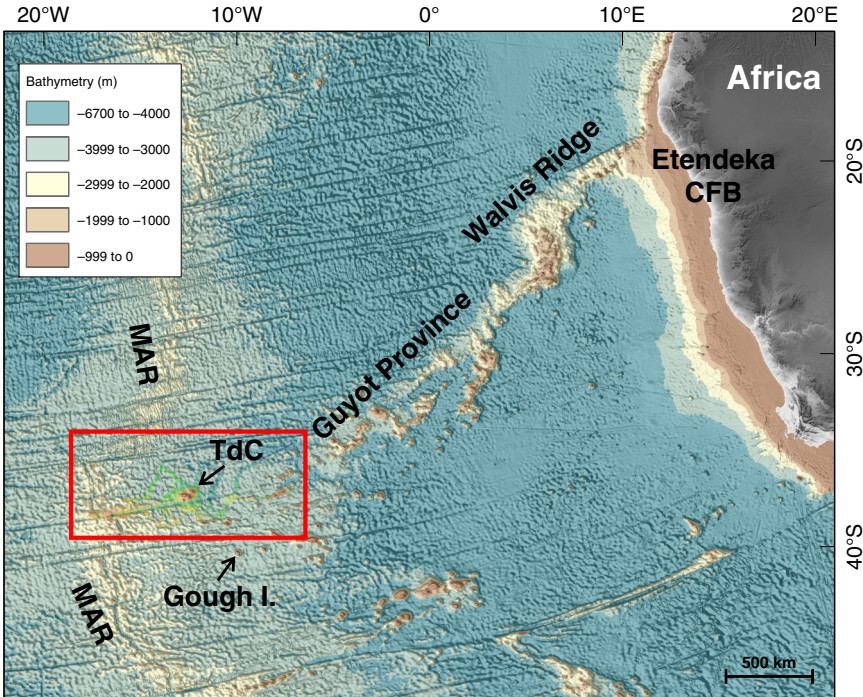

**Fig. 1 Bathymetry of the eastern South Atlantic.** GMRT v.3.2 bathymetry[42] of the SE Atlantic with the Walvis Ridge and the study area around Tristan da Cunha (red box). CFB continental flood basalt province; I. Island; MAR Mid-Atlantic Ridge; TdC Tristan da Cunha.

data provide an unprecedented view of an area that was previously largely unmapped with MBES techniques. Amongst other Geographic Information System (GIS) tools, the Benthic Terrain Modeler (BTM)[34] is used to describe the volcanic and sedimentary morphology of the seafloor. Differences in signal penetration and reflectivity produce characteristic patterns in sub-bottom profiler data that enable us to interpret the seafloor fabric related to seafloor spreading, as well as the presence of thick pelagic sediments or volcaniclastic material. Despite the lack of ground truth from geological samples, we are able to document clear evidence for relatively recent seafloor eruptions. We discuss the implications of these observations for plume magmatism processes.

## Results

**Data coverage.** The regional distribution of our new MBES bathymetry and backscatter data at 30 m grid resolution is shown in Fig. 2. All but the very shallowest seafloor is covered within a radius of ~45 km around the archipelago. Further afield, near-complete coverage was also achieved at the intersection of the TTFZ with the MAR. More scattered data were collected along regional tracks between seismological and electromagnetic ocean-bottom stations and on transits to the area[35,36]. We describe features of the seafloor in three-specific areas (see Figs. 2–8).

**Central area.** On the main island's submarine flanks, the contour at 2000 meters below sea level (mbsl) is almost circular with a diameter of ~20 km (Fig. 3a). At this depth, a saddle to the southwest of the island connects the edifice to a plateau bearing the neighboring islands. Away from this saddle, the TdC edifice deepens to 3600 mbsl at distances of ~25 km from the island. Sedimentary aprons extend outwards a further 40 km or more. The edifice has steep (>15°) slopes, within the range for oceanic shield volcanoes[16], that are characterized by radial ridges and

channels as well as the presence of numerous parasitic cones. High seabed backscatter intensities, as shown in Fig. 3b (see Supplementary Figs. 2 and 3 for gray-scale images), suggest a relatively coarse, rough, and/or hard seafloor, which most likely consists of volcanic material. Further out, the bathymetric rise of the edifice is dominated by sedimentary aprons covering pre-existing abyssal hill fabric. Here, lower backscatter intensities indicate greater absorption of acoustic energy, consistent with the presence of pelagic or hemipelagic sediment and/or very fine-grained volcaniclastics. In the NW sector, a set of lobate bed-forms, identifiable in backscatter and rugosity (a derivate of the bathymetry, Fig. 3c), can be interpreted as deposits of past slope failure events (see also Holcomb and Searle[37]) resulting from one or more sector collapse events[14] (Supplementary Fig. 4).

Inaccessible Island (Fig. 3; 37°19′S 12°41′W, ~14 km$^2$) is located on a W–E elongated insular shelf (~200 km$^2$) with a water depth of probably <400 m. The island likely evolved from two or more individual volcanoes[38,39]. The shelf's flanks, like those of TdC, are steep (>15°) and marked by radial ridges, channels, and parasitic cones, and are dominated by high backscatter intensities. At its base, the Inaccessible Island edifice covers an area of about 25 × 40 km.

The shallow shelf around Inaccessible Island connects south-eastwards to a ~25 km$^2$ shelf around Nightingale Island, where water depths are less than 150 m. The edifice of Nightingale Island (Fig. 3; 37°26′S 12°29′W, ~2.6 km$^2$) has a basal diameter of ~17 km, encompassing steep slopes with angles at >15° very similar to those of TdC and Inaccessible Island.

The new datasets reveal previously unknown features, including two submarine volcanoes to the southwest and southeast/east of Nightingale Island, named Rockhopper and Isolde Seamounts (Fig. 3). Rockhopper Seamount (37°34′S 12°45′W) has a basal diameter of about 20–22 km and rises to a depth of 470 mbsl at its summit, almost 2500 m above the adjacent seafloor (Fig. 4a, b). The

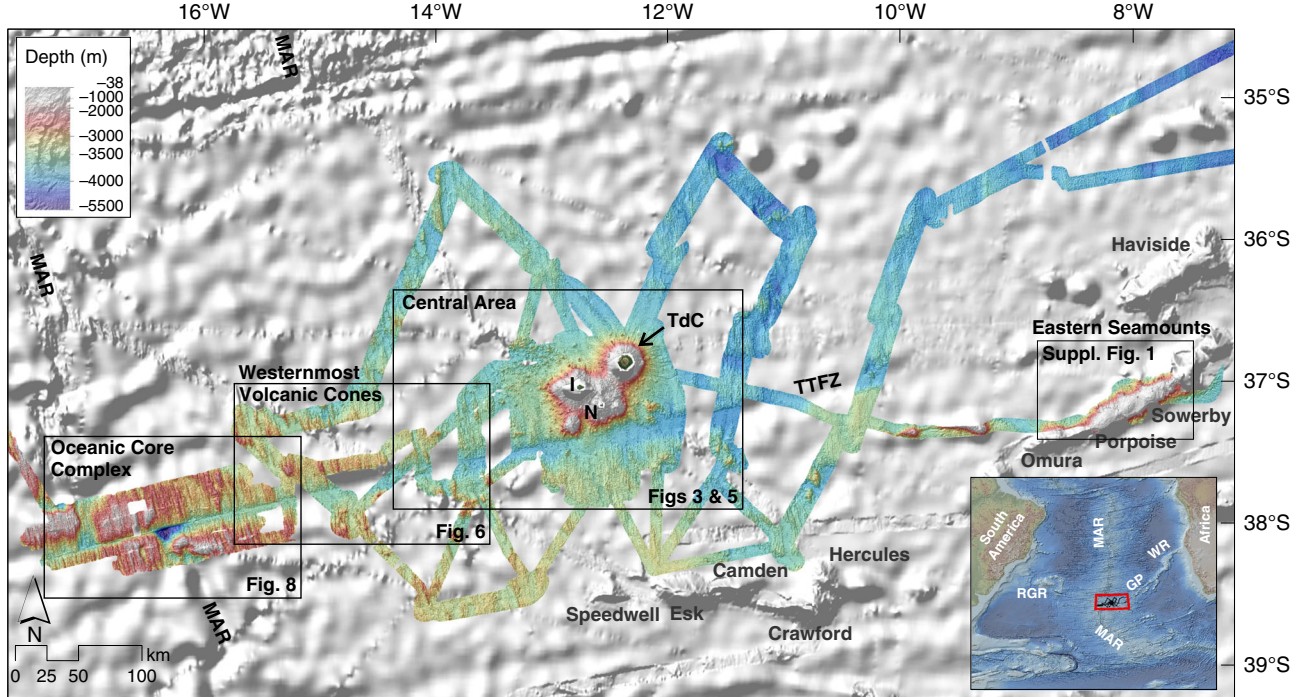

**Fig. 2 Overview around Tristan da Cunha and fracture zone.** Regional overview of new multibeam echosounder bathymetry in the vicinity of Tristan da Cunha and along the Tristan da Cunha transform fault and fracture zone system. Compilation includes data from NGDC[69] and UKHO[70]. The hillshade is illuminated by an azimuth of 315°. Boxes indicate regions shown in more detail in Figs. 3–7. The maps for the eastern seamounts are shown in Supplementary Fig. 1. Seabed backscatter maps are shown in Supplementary Fig. 2. GP Guyot Province; I Inaccessible Island; N Nightingale Island; MAR Mid-Atlantic Ridge; RGR Rio Grande Rise; TdC Tristan da Cunha; TTFZ Tristan da Cunha Transform Fault And Fracture Zone System; WR Walvis Ridge.

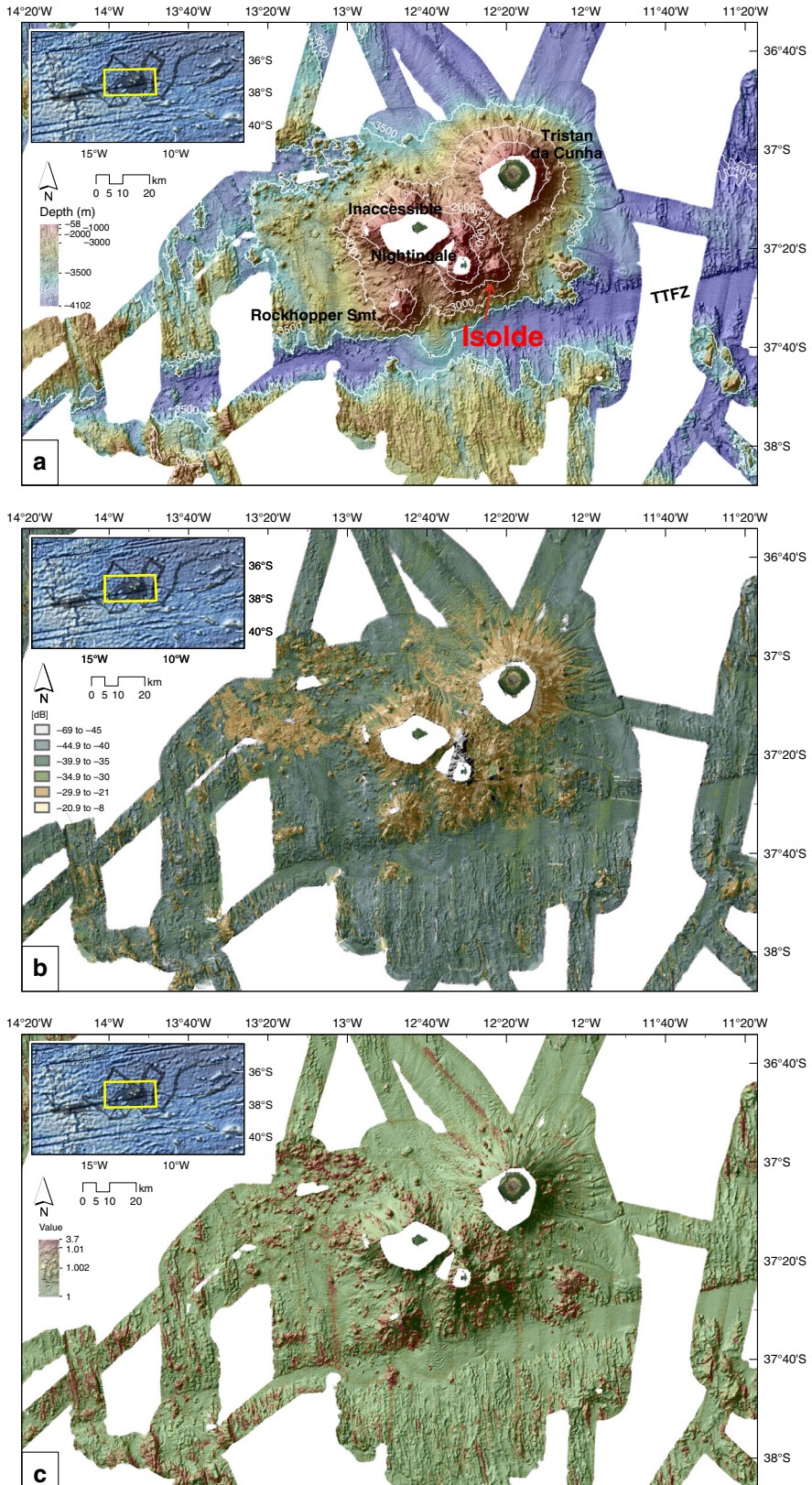

**Fig. 3 Central area bathymetry, backscatter, rugosity. a** Bathymetry for the central area. The hillshade is illuminated by an azimuth of 315°. The newly discovered Isolde and Rockhopper Seamounts (Smt.) lie south and southwest of Tristan da Cunha. **b** Seabed backscatter classification based on histogram natural breaks [dB]. Unclassified greyscale version in Supplementary Fig. 3a. **c** Rugosity after DuPreez[63]. TTFZ Tristan da Cunha Transform Fault And Fracture Zone System.

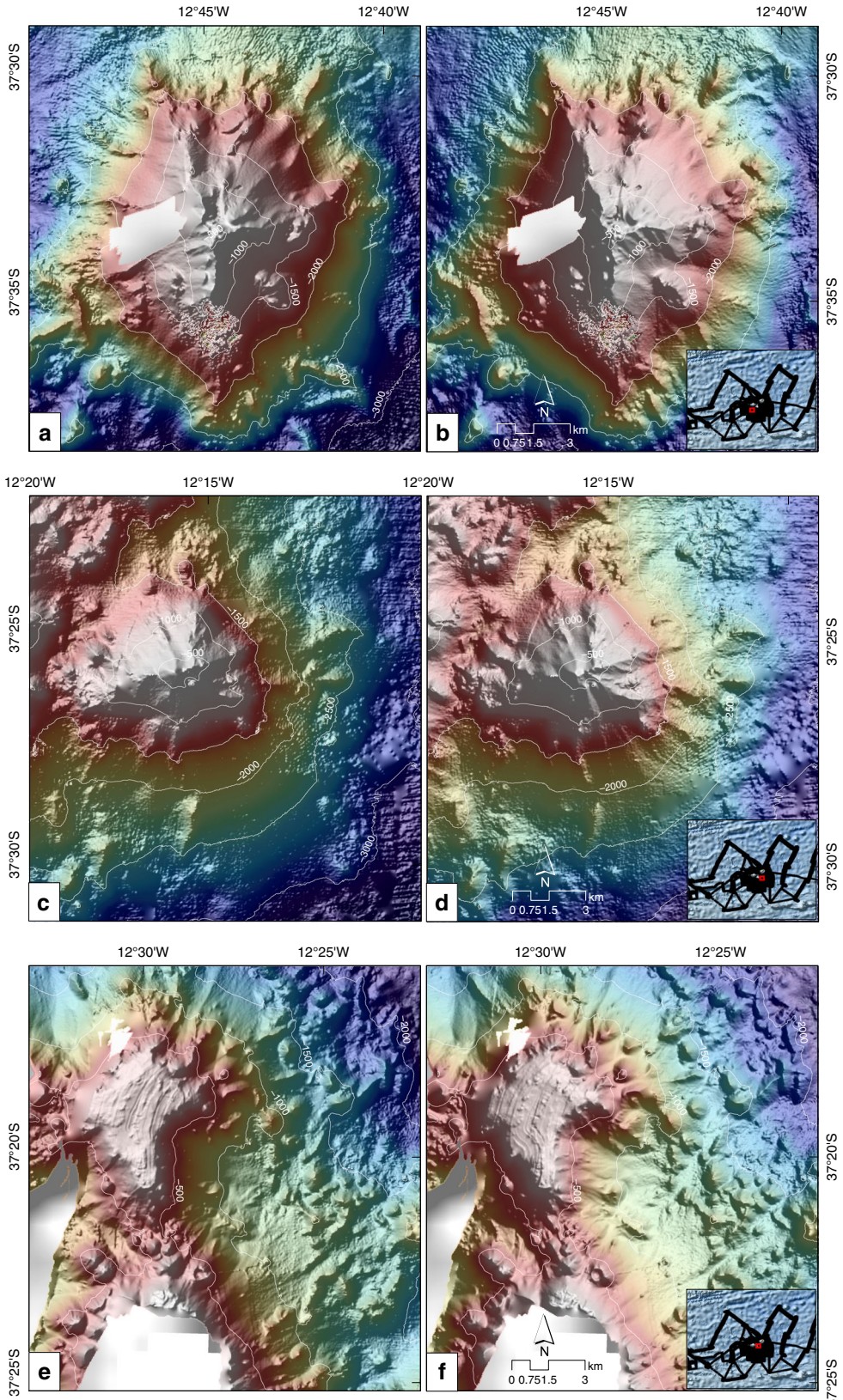

**Fig. 4 Central area bathymetry close-ups. a**, **b** Bathymetry of the Rockhopper Seamount. The hillshade is illuminated by an azimuth of 315 and 45°, respectively. **c**, **d** Bathymetry of the Isolde Seamount. The hillshade is illuminated by an azimuth of 315 and 45°, respectively. **e**, **f** Bathymetry of the small plateau to the north of Nightingale Island and east of Inaccessible Island. The hillshade is illuminated by an azimuth of 315 and 45°, respectively.

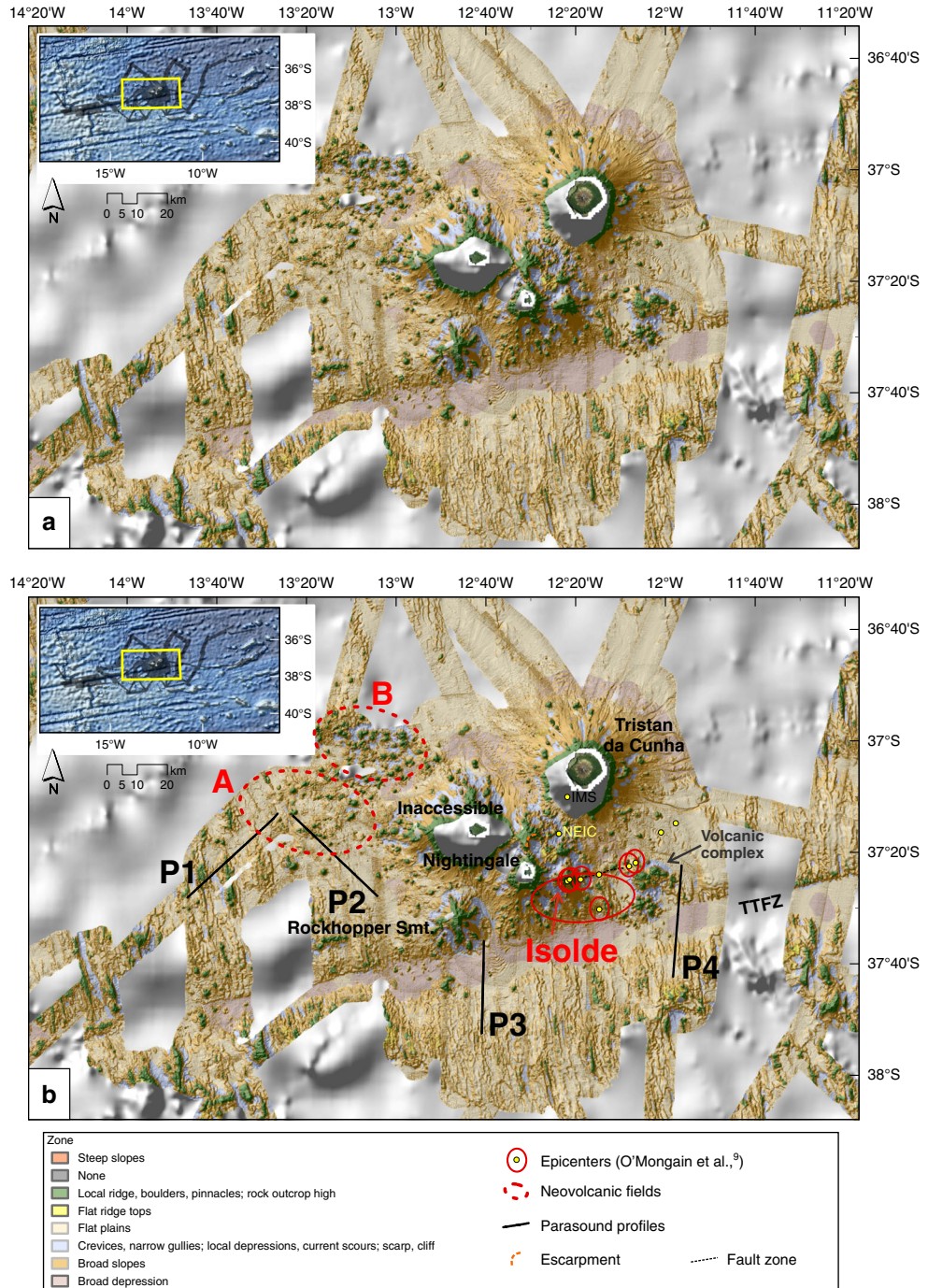

**Fig. 5 Benthic Terrain Model of the central area.** Benthic Terrain Model. Yellow dots with red ellipses to the SE of Nightingale Island mark local earthquake epicenters associated with submarine volcanic activity in 2004[9], and their uncertainties. Areas A and B outline relatively young volcanic fields with monogenetic cones and lava flows. P1 to P4 mark locations of sub-bottom profiler (Parasound) profiles. Smt. Seamount; TTFZ Tristan da Cunha Transform Fault And Fracture Zone System.

summit forms an NNW–SSE elongated ridge that is almost parallel to the region's abyssal hill fabric. A second, less prominent WSW–ENE striking ridge runs almost parallel to the TTFZ (Fig. 5). With >20°, the seamounts' slopes are considerably steeper than the slopes of the islands, but still within the range of oceanic shield volcanoes[16]. A sedimentary fan, characterized by low backscatter values (<−30 dB), is imaged to the southeast of Rockhopper Seamount (Fig. 3). Isolde Seamount (37°26′S 12°20′W) has a basal diameter of 10 km and rises with slopes of ~18° to a depth of ~300 mbsl, about 1200 m above the seafloor in the saddle between it and

Nightingale Island (Fig. 4c, d). The summit lies on a ridge that is elongated ENE, almost parallel to the TTFZ. The data reveal the presence of a ~500 m-wide depression whose floor lies at ~550 mbsl, which we interpret as a summit crater. A more detailed survey would be necessary to test this interpretation. In addition to the new named seamounts, a small eroded plateau (4 × 4 km, with a basal diameter of 6 km) is mapped in water depths of 200–220 mbsl to the north of Nightingale Island and east of Inaccessible Island (37° 19′S 12°30′W) (Fig. 4e, f). The plateau shoals to about ~160 mbsl at its southernmost part. The plateau's SE flank is marked by steep

arcuate scarps that may be evidence of slope failure events or volcanic eruptions (Fig. 5). Another small edifice is mapped just to the north of Isolde Seamount (37°34′S 12°28′W) (Fig. 3). Its summit is elongated ENE and rises to about 850 mbsl. Its similarly elongated base measures 6 × 4 km.

To the east of Isolde Seamount, we mapped several volcanic cones and a larger volcanic complex, spatially related to earthquake epicenters of the 2004 volcanic unrest[9] (Fig. 5). We interpret an escarpment that extends eastwards from Isolde Seamount as the trace of a fault (Fig. 3). In addition to these features, our data reveal numerous small secondary volcanic cones on the flanks of the islands and seamounts. Northwest of Inaccessible Island, areas A and B in Fig. 5 show evidence for volcanic activity in the form of monogenetic volcanic cones, knolls, and lava flows. Whereas area B is characterized by the abundance of cones, area A is dominantly covered by lava flows, which are identifiable from higher backscatter intensities (Fig. 3b) and from the sub-bottom profile P2 (see below). Further west, between 14°20′W and 13°20′W, the seafloor is marked by scattered small volcanic cones. Only a very few of these cones show summit crater structures. Some of the cones appear to have coalesced to form ridges.

Our data reveal no evidence for the presence of major fault zones other than the TTFZ, which runs ENE–WSW across the deep seafloor south of the archipelago. There is little evidence for intraplate volcanic activity within the TTFZ itself, or further south of it.

**Area of the westernmost volcanic cones**. Figure 6 details the area of the TTFZ halfway between TdC and the MAR. This area is not yet fully covered by shipboard bathymetry, but the existing data allow some preliminary observations. The bathymetric data (Fig. 6a) show undisturbed abyssal hill fabric away from the fracture zone, consisting of NNW–SSE oriented hills and ridges formed along the MAR over the past 5–20 Myrs[21,22]. The BTM clearly illustrates this pre-existing normal fabric (Fig. 6b). Scattered small volcanic cones and W–E oriented ridges are visible along and near to the TTFZ, with decreasing number and frequency westward. Evidence for volcanic activity seems to extend out to 250 km westwards of the TdC island group (as far as ~14° 48′W), where we mapped a small volcanic cone on top of the abyssal hills of ~10 Myr-old oceanic crust[21,22] just 130 km east of the MAR. The average strike of the abyssal hills around this cone (~178°) differs slightly from the regional pattern (~165°).

**Volume of volcanic cones**. As shown above, numerous cone-shaped morphologies were mapped and interpreted as volcanic cones around and to the west of the TdC archipelago. Despite limitations in data coverage, we were able to identify and quantify the size and volume of potential volcanic cones (Fig. 7). In total, we identified 430 cones, including Rockhopper and Isolde Seamounts, covering an area of >1300 km² and comprising a volume of ~200 km³ of magmatic extrusive rocks (see Supplementary Table 1). Over 95% of the mapped cones have a diameter of less than 3 km, with most diameters in the range 1–2 km. Therefore, the majority of the cones cover a basal area of <5 km² with an overall mean basal area of 1.8 km². Net cone volumes are generally <0.7 km³ with an overall mean net volume of 0.5 km³.

**Mid-Atlantic Ridge axis and nearby region**. The slow-spreading southern MAR (half-spreading rate 1.5 cm/yr[22]) is strongly segmented by transform faults that continue as fracture zones into the surrounding plate interiors. One of the largest ridge-crest offsets in the South Atlantic Ocean is a 250 km long transform fault bounding the TdC spreading corridor to the north, observable at roughly

35° S (Figs. 1 and 2). To the south, the segment is bounded by the 25 km long TdC transform fault, which then continues as the TdC Fracture Zone further eastward. Both together form the TTFZ[40].

The intersection of the MAR with the TTFZ is illustrated in Fig. 8. Water depths are generally shallower than 3000 m (Fig. 8a). However, in the median valley of the MAR rift valley and the valleys marking the transform fault and fracture zone water depths are typically between 3000 and 4000 m, but reach more than 4800 m in the nodal basin. Southeast of the eastern ridge–transform intersection at 38°25′S, 16°20′W, a newly discovered domal high rises to <2000 mbsl (Fig. 8a). The dome surface displays several ENE–WSW oriented, spreading-parallel striations (Fig. 8b), which are also well-imaged in the backscatter data (Fig. 8c). The striations reach lengths of up to 17 km and are closely spaced, giving the surface a corrugated appearance. Corrugated surfaces like this were first interpreted by Cann et al.[41] as detachment faults bounded by active mid-ocean ridge and transform faults (at the so-called inside-corners of ridge–transform intersections), and termed oceanic core complexes (OCC). By accommodating plate divergence in the absence of melt, OCCs act to exhume deep crustal and uppermost mantle rocks, leaving them exposed at the seafloor. Here, unusually, the OCC is found in the outside corner of the ridge–transform, whilst the inside corner displays normal abyssal hill fabric. Further east, the bathymetry reveals almost regular abyssal hills with NNW–SSE oriented ridges and valleys, indicating the absence of OCCs in the past on this flank of the MAR (Figs. 2, 6, and 8). North of the TTFZ, the flanks of the MAR display a regular pattern of long abyssal hills without any evidence for an OCC at either corner of the ridge–transform intersection.

**The Tristan da Cunha Transform Fault And Fracture Zone System**. The TTFZ is ~1 km wide near the MAR and widens to ~3 km further east, becoming even broader in areas with sedimentary or basaltic cover. Near the MAR axis, water depths along the active transform fault exceed 4800 m within the nodal basin, whereas in the fracture zone valley water depth varies from 3400 m close to the MAR to 3750 m further east (Fig. 2, Supplementary Fig. 5). East of the TdC island group, the fracture zone inner valley deepens to 4000 mbsl, shoaling again eastwards to 3750 mbsl. Differences between the satellite-derived depths in the global dataset GMRT v3.2[42] and those in our high-resolution survey reach up to 200–250 m (Supplementary Fig. 5). Volcanic edifices within the TTFZ rise up to 400 m above the neighboring seafloor. The presence of the Rockhopper sedimentary fan makes the seafloor to the south of the islands ~350 m shallower than in the surrounding areas of the fracture zone (Supplementary Fig. 5).

**Sub-bottom profiler data (Teledyne ATLAS Parasound P70)**. Sub-bottom profiler data are used to characterize the acoustic fabric of the seafloor and shallow subseafloor. They can be used to distinguish between buried and exposed seafloor fabric related to seafloor spreading, the presence and internal structure of pelagic sediment cover, and areas covered by relatively young volcanic products.

Parasound profile P1 (Fig. 9a) is oriented SW–NE about 70 km west of Inaccessible Island (Fig. 5b). The profile shows a row of relatively old ridges (abyssal hills) formed before ~15 Ma by seafloor spreading at the MAR[21,22]. The ridge flanks and the valleys and plains between them are partly covered by thin (<50 m) sediments that are characterized by low seabed backscatter values (<−35 dB). At its northeastern end, profile P1 images relatively young volcanic overprint of the seafloor

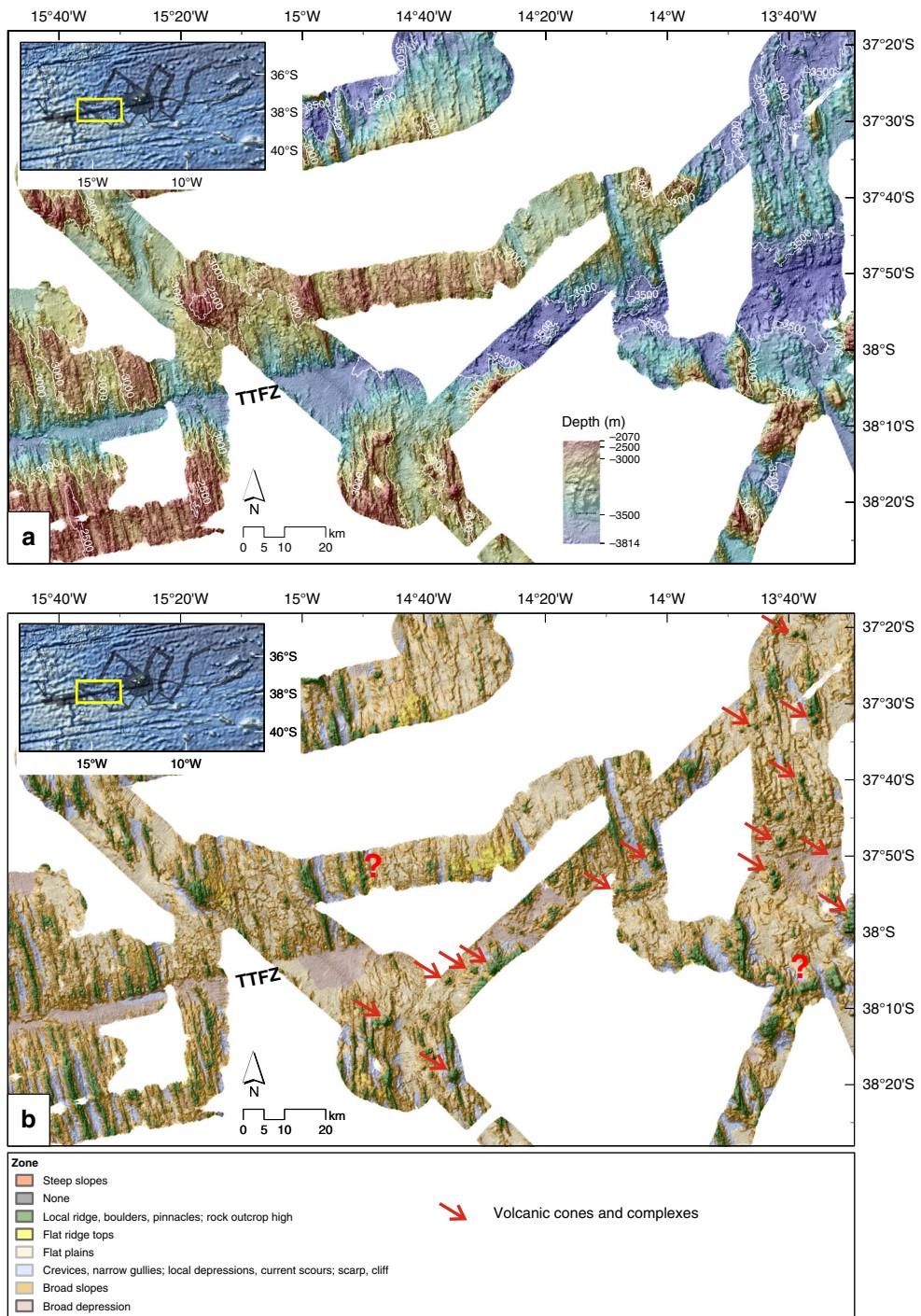

**Fig. 6 Bathymetry and Benthic Terrain Model of the western volcanic cones. a** Bathymetry for the area of the western volcanic cones. The hillshade is illuminated by an azimuth of 315°. **b** Benthic Terrain Model. A number of circular features and W–E orientated ridges can be identified that are interpreted as relatively young volcanic cones and complexes (red arrows). TTFZ Tristan da Cunha Transform Fault And Fracture Zone System. Unclassified greyscale seabed backscatter is shown in Supplementary Fig. 3b.

(compare to Figs. 3b and 5, area A with high seabed backscatter values > −25 dB).

The seafloor to the west of Inaccessible Island is imaged in profile P2 (Fig. 9b). The southeastern part of the profile images undisturbed strata accumulated by pelagic sedimentation (resolved thickness about 50 m). Areas with sedimentary cover like this are characterized by low backscatter values (<−35 dB). Northwestwards, the profile shows a more disrupted, hummocky pattern. At the transition to these disturbed strata, the reduced

amplitudes of the deeper sedimentary reflections may record the masking effect of a thin volcanic cover with increased impedance contrast (see inset in Fig. 9b). Diffraction hyperbolas further west indicate relatively steep or rough sharp-peaked volcanic cones and complexes within the area labeled "A" in Fig. 5b. This area is also characterized by high backscatter values. No sedimentary layers are observed above or beneath these volcanic features, which may therefore comprise a relatively thick volcanic layer. The Parasound profile clearly shows that the volcanic cover in the

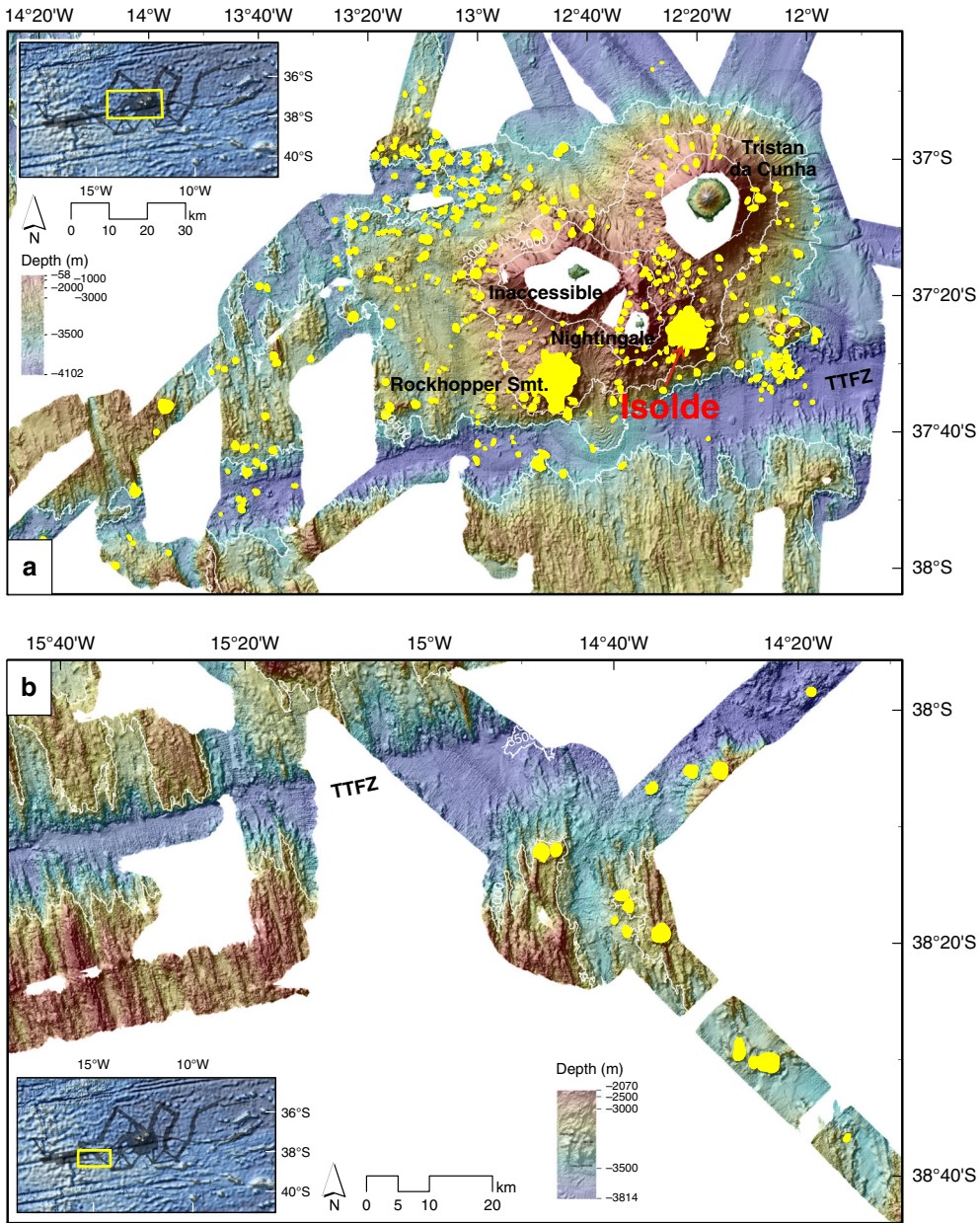

**Fig. 7 Thematic maps of identified volcanic cones. a** Bathymetry for the central area with identified volcanic cones. **b** Bathymetry for the western area with identified volcanic cones. The hillshade is illuminated by an azimuth of 315°. Smt. Seamount; TTFZ Tristan da Cunha Transform Fault And Fracture Zone System.

northwest is younger than the undisturbed sedimentary layer in the southeast. Assuming this 50 m-thick layer to have accumulated onto the underlying ~15 Myr-old crust at a constant rate yields a regional background pelagic sedimentation rate of 3.33 m/Myr. Using this rate together with the observation that no sediments drape the volcanic cover in the northwest, and the expectation that the minimum unresolvable thickness of sediment in Parasound data is ~0.2 m[43] (see "Methods"), then the maximum age of volcanic cover is 0.06 Ma.

Parasound Profile P3 (Fig. 9c) runs across the foot of a submarine sedimentary fan between Rockhopper Smt. and Nightingale Island (Fig. 5b). The fan covers an area of about 30 × 20 km, crossing the floor and one wall of the TTFZ valley. The opposing wall of the valley is characterized by undisturbed pelagic sediments that are about 50–100 m thick. Especially at the southern flank, it appears that many faults occur within the

sedimentary layers. However, a closer look indicates that most of these features are related to sedimentary waves caused by contourite deposition or represent artefacts caused by trace scaling (Fig. 9c).

Parasound profile P4 (Fig. 9d) illustrates a section across the part of the TTFZ that lies southeast of TdC. The apparent width of the fracture zone valley here is ~10–12 km. The valley flanks lie at between 3600 and 3800 m water depth, whereas the valley floor is 3800–3900 mbsl deep. According to the sub-bottom profiler data, both the fracture zone and its flanks are covered by sediments. Sediment thickness over the valley flanks is less (50–100 m) than in the valley floor (>150 m). Backscatter intensity values are generally less than −30 dB. Diffractions at the northern shoulder indicate the presence of volcanic over-printing products at the seafloor. As in profile P3, apparent fault-like features across the southern flank are attributable to trace

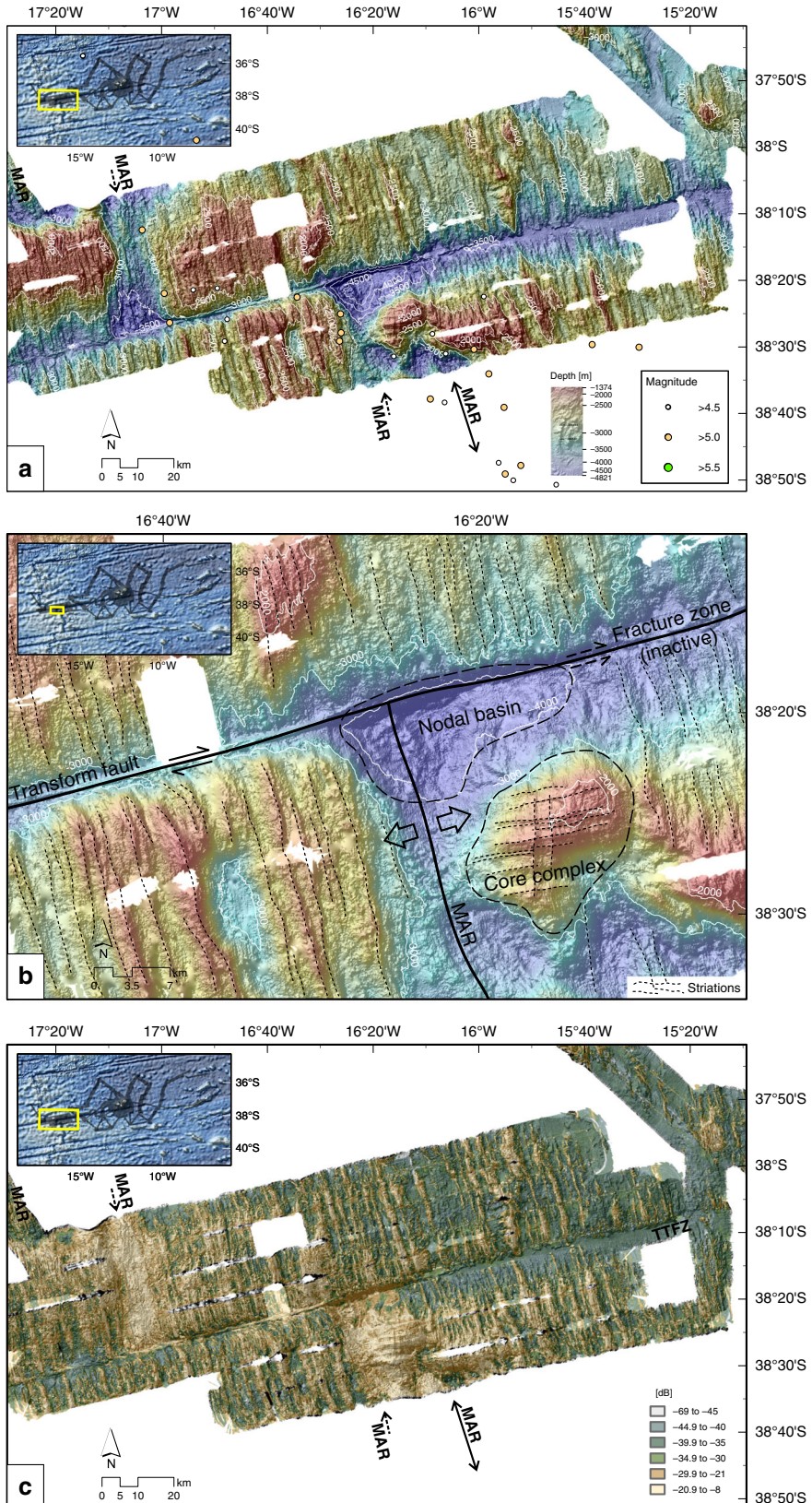

**Fig. 8 Bathymetry and seabed backscatter of Mid-Atlantic Ridge area. a** Bathymetry for the area close to the Mid-Atlantic Ridge. Dots mark earthquake epicenters from teleseismic observations[52]. The hillshade is illuminated by an azimuth of 315°. **b** Close-up of the oceanic core complex. **c** Seabed backscatter data illustrating abyssal hills and oceanic core complex. Unclassified greyscale version is shown in Supplementary Fig. 3c. MAR Mid-Atlantic Ridge; TTFZ Tristan da Cunha Transform Fault And Fracture Zone System.

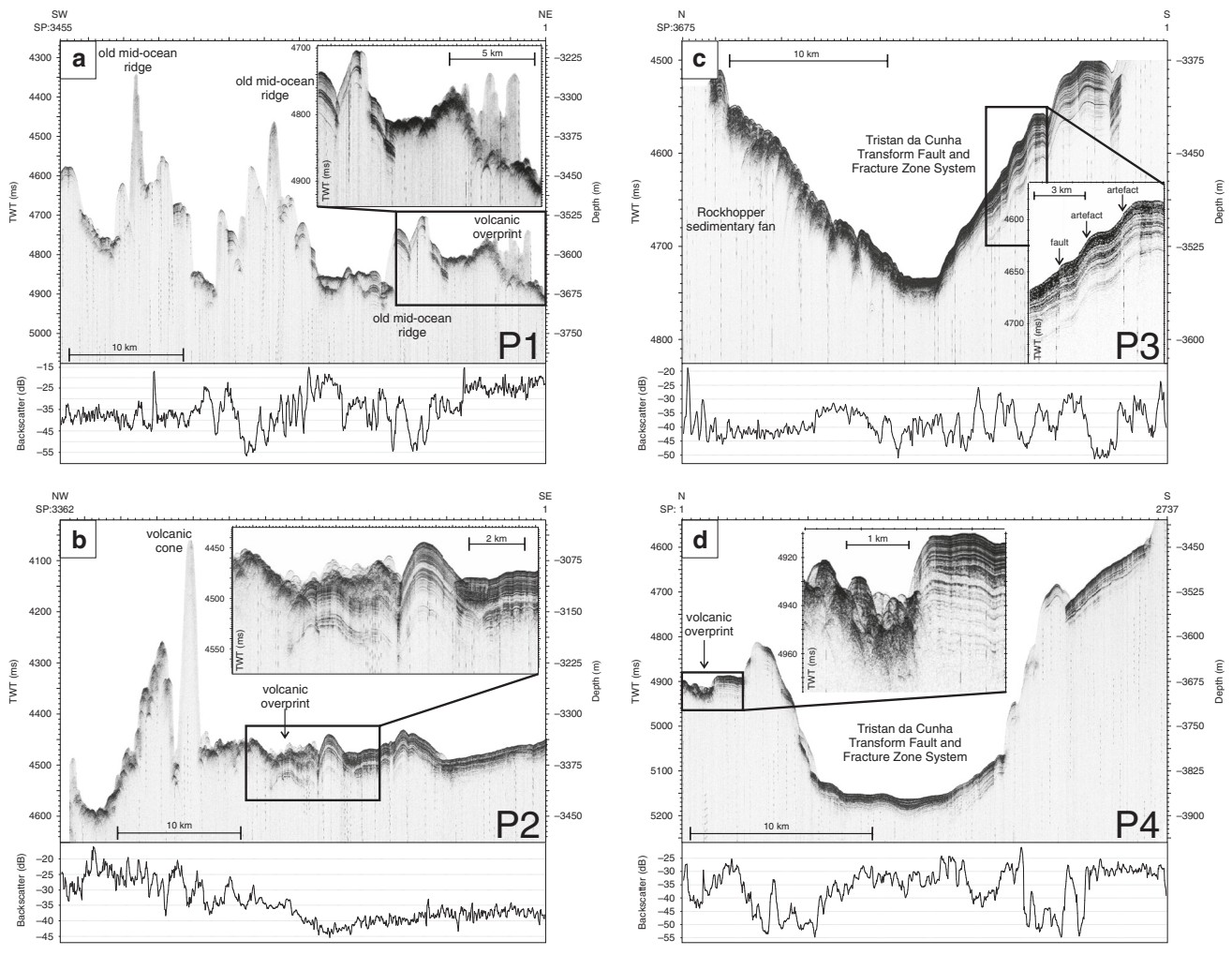

**Fig. 9 Sub-bottom profiler sections.** Sub-bottom profiler (Parasound) profiles (see Fig. 5b for location). Seabed backscatter intensities are plotted below each profile. **a** Profile P1 across the oceanic seafloor to the west of Tristan da Cunha. **b** Profile P2 across the southeastern border of volcanic field A. **c** Profile P3 across the Tristan da Cunha Transform Fault And Fracture Zone System and the southern part of the submarine fan southwest of Nightingale Island. **d** Profile P4 across the Tristan da Cunha Transform Fault And Fracture Zone System southeast of Tristan da Cunha. Most of the apparent steep-to-vertical offsets in the sub-bottom profiler data are artefacts caused by trace scaling.

scaling. Small step-like features at the seafloor (e.g., southern flank in Fig. 9d) could be related to bottom current deposition.

## Discussion

Detailed studies from the region around Hawai'i have shown that oceanic intraplate volcanoes apparently evolve through different magmatic stages in response to temporal changes in magma sources that reflect evolving degrees of partial melting and magma production[44,45]. This evolution probably in turn reflects the motion of the lithosphere across the melting anomaly, most likely a deep mantle plume[46]. In the case of Hawai'i, Loihi Seamount is the youngest volcano southeast of the main island. Loihi is located above a seismic low velocity anomaly in the upper mantle[5]. Applying the concept of Clague and Sherrod[47] developed for Hawai'i, the mapped volcanic edifices close to TdC are in a preshield phase as Loihi is. The mapped volcanic cones and knolls northwest and west of the TdC might be in the so-called basal phase of the seamount development scheme developed by Schnur and Gilbert[48].

Small volcanoes representing an early volcanic stage have also been found near the large volcanoes of Pitcairn and Tahiti in the Pacific[4,49]. These relatively young (344–3 ka[49]) submarine structures range between volcanic cones up to 500 m high and much larger edifices with heights of several kilometers. These volcanoes erupted mainly highly differentiated magmas, melts that were derived exclusively from the most trace-element enriched, isotopically extreme mantle component. This component has the lowest melting temperature and so should form the first products of melting of a new batch of mantle[4] during the earliest stages of hotspot activity in any new region.

Hagen et al.[2] reported on large areas of submarine volcanism north and west of Easter Island. Lava flows and small volcanoes were mapped by SeaMARC II sidescan sonar data in which high backscatter values were interpreted as evidence for relatively recent submarine volcanic activity. Fretzdorff et al.[3] studied enriched and depleted tholeiites dredged from the young Umu volcanic field west of Easter Island. Their modeling of partial melting and magma mixing points to the involvement of both enriched and depleted plume components, both of which could be related to the Easter plume source.

While the numerous up to 400 m high cones west of TdC and Inaccessible islands may also represent analogous features to the small submarine volcanoes at the Pitcairn and Easter hotspots, the lack of samples means their geochemistry remains unknown. Nevertheless, we speculate that the two areas west-northwest of

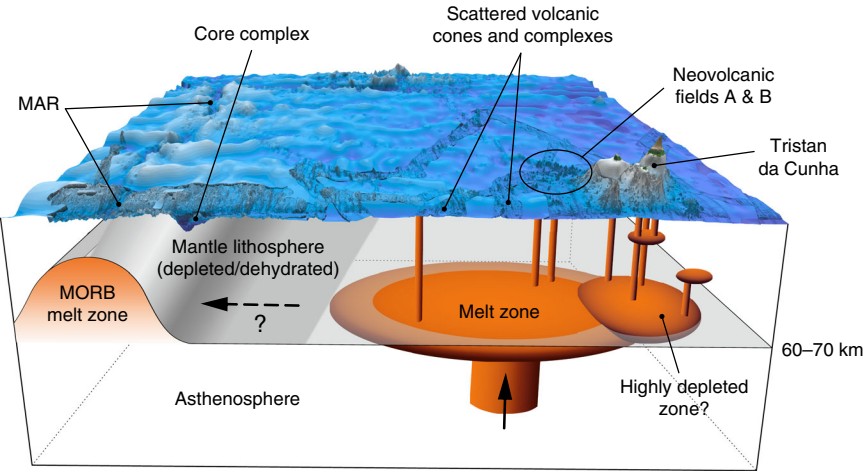

**Fig. 10 Schematic model of Tristan da Cunha mantle plume.** 3D sketch of the lithosphere and potential mantle plume location SW of Tristan da Cunha. The model is based on Geissler et al.[21], Humphreys and Niu[50], and Schloemer et al.[31].

Inaccessible Island, each about 20 × 40 km across, with cones and lava flows causing high backscatter in comparison to surrounding sediment-covered undisturbed seafloor, might represent the incubators of future large seamounts or even islands. Perhaps, consistent with this, Schloemer et al.[32] detected earthquake activity below the western part of area B, possibly indicating active tectono-magmatic processes.

In Fig. 10, we present a model for the plume-lithosphere system at TdC based on Geissler et al.[21], Humphreys and Niu[50] and Schloemer et al.[31]. Following Schloemer et al.[31], the center of the TdC mantle plume is situated close to Inaccessible Island assuming that TdC was formed at 1.3 Ma[6] directly above the plume center and subsequently migrated away northeastward with the African Plate at 3.2 cm/yr[51]. Consistent with this, Schloemer et al.[31] imaged a low-P-wave seismic velocity anomaly (their so-called Tristan conduit), with its edge below Inaccessible Island, further southwest. The projection of this conduit to the seafloor intersects the area of numerous small volcanic edifices. Slower average S-wave seismic velocities in the upper mantle were also observed in this area by Bonadio et al.[33] by analyzing teleseismic surface waves. Among the range of possible causes for relatively recent tectono-magmatic activity between TdC and the MAR, these seismic anomalies independently support the presence of warm and melt-rich (<1%[33]) upper mantle. According to Baba et al.[29] and Geissler et al.[21], the regional thickness of the oceanic lithosphere is about 60–70 km. Another high-P-wave velocity body was inferred by Schloemer et al.[31] in the lower lithosphere directly beneath the islands that could be related to the presence of a highly depleted and dehydrated zone.

The westwards regional extent of anomalous upper mantle is not clearly resolvable with the seismological data[31,33]. We mapped volcanic cones in our bathymetric dataset, noting a westward decrease in the frequency of their occurrence (Fig. 7). From the available bathymetry, cones appear to be absent westward of a point 250 km WSW of TdC and 130 km east of the MAR (Fig. 7). Global[52] and regional[32] seismicity data show local earthquake activity close to the westernmost mapped volcanic cones.

Humphris et al.[53] reported a geochemical anomaly amongst sparsely sampled data along the MAR west of TdC, reporting enriched basalts but no clearly plume type basalt. Humphris et al.[53] interpreted these results as an indication of poor and waning communication between TdC plume mantle and normal mantle beneath the MAR in a region where the two were previously better mixed. Without dense sampling of rocks along this portion of the MAR, including the mapped OCC, we cannot

provide any new constraints to this problem. The absence of an OCC at the inside corner of the northern ridge–transform intersection on the TTFZ suggests melt supply there. This melt may be slightly enhanced by the presence of Tristan plume mantle. To the south, the presence of a relatively recently formed OCC suggests a lower melt supply, whilst the absence of fossil OCCs suggests that this condition may be a relatively recent development. OCCs occur at slow-spreading ridges[54] and reflect a melt supply rate that is insufficient to support the symmetrical formation of a thick magmatic crust, resulting in an asymmetric spreading, with relatively normal basaltic crust emplaced on the opposite ridge flank[54–58]. OCCs are often associated with ridge–transform intersections but may also occur near smaller offsets or along entire segments, as is observed at the MAR at 13–15°N[59]. OCCs are composed of plutonic rocks, mostly gabbros associated with serpentinized peridotites[56]. They are, in all cases, products of cooler spreading regimes with lower melt supplies at the ridge axis. The OCC mapped here suggests the absence or intermittency of plume influence on the MAR south of the TTFZ. Taking the lengths of the longest striations as 17 km and an average half-spreading rate of 1.5 cm/yr[22] for this part of the ridge, we can estimate that the formation of the OCC took about 1.1 Myrs. Since the striations end actually before the MAR axis, the exhumation could have started before 1.1 Ma.

The last destructive onshore volcanic eruption on TdC took place in 1961–1962, necessitating evacuation of the island's main settlement[7]. In July 2004, tremors were felt on the island, raising concerns about another eruption. However, there were no signs of increased onshore thermal activity or ground deformation, such as increased rock fall activity or disruption of man-made structures, on the island[8]. Subsequently, pumice was discovered by cray fishers on Sandy Point Beach on the island's east coast[8]. Taking the direction of prevailing currents into consideration, the locations of the pumice discoveries suggest its source lay south or southwest of TdC.

Subsequent analysis of seismological data from two stations on the island showed a dramatic increase in earthquake activity beginning on 29 July 2004, with more than 2000 local earthquakes detected between July and December 2004[9]. The initial earthquakes showed a consistent increase in amplitude with time, culminating in four earthquakes with magnitudes of 4.7–4.8 mb on 29–30 July. Within their relatively large location errors, all of the events were located between 37 and 53 km S and SSE of TdC (see Fig. 5b).

From Fig. 5b, it is obvious that the epicenters of the July 2004 earthquakes cluster close to Isolde Seamount. Together with the

evidence from the pumice finds, this suggests that the 2004 submarine volcanic eruption may have occurred at Isolde Seamount. The Isolde volcano rises into water depths of ~300 m, well within the range at which submarine eruptions can produce pumice[60,61]. A few of the recorded seismic events occurred further east than Isolde. Considering the ~W–E elongation of Isolde Seamount and the presence of further small volcanic cones to the east of it (see Figs. 3, 5, and 7), these earthquakes indicate that the 2004 seismic crisis activated a fault zone east of Isolde (Fig. 5b). Schloemer et al.[32] located two earthquakes in 2012 in the close vicinity of Isolde Seamount, at focal depths of 5–15 km. These events suggest that the plumbing system of the volcano may still be active. Deeper earthquakes (up to 30 km depth) were located closer to the TTFZ by Schloemer et al.[32].

**Conclusions**. MBES bathymetry, seabed backscatter, and sub-bottom profiler data around TdC show evidence for relatively recent submarine volcanic activity. The GIS based BTM and backscatter analysis were used for mapping volcanically altered oceanic terrain based on objective criteria. Combining our findings with seismological results, we interpret that these relatively young volcanic cones are being fed by TdC mantle plume material, but this will have to be confirmed by future sampling. Our data further challenge the assumption that the present-day Walvis Ridge-TdC hotspot is located directly below TdC. Instead, the hydroacoustic data support a hotspot location west of the island, consistent with recently published geophysical data. The newly discovered Isolde Seamount south of TdC was likely the location of a submarine eruption in 2004. An OCC mapped at the ridge–transform intersection west of TdC indicates a relatively recent reduction in melt supply via plume–ridge interaction south of the TTFZ.

Future rock sampling of the newly discovered submarine volcanoes will be required to test these ideas and better characterize the geochemical composition of the Tristan mantle plume. Combining such geochemical information with the existing data base along the Walvis Ridge will improve hotspot and mantle melting models for the Walvis Ridge hotspot track and mantle plume interaction with the MAR.

## Methods

**Multibeam echosounder data**. Hydroacoustic data were acquired by a hull-mounted deep-water KONGSBERG EM120 MBES with 191 beams during cruise MSM20/2[36] and by the upgraded KONGSBERG EM122 with up to 864 soundings in high-density and dual-swath mode during cruise MSM24[35]. The ship's speed varied from 13 kn during the transits to 8 kn for dedicated bathymetry surveys around TdC, the TTFZ and the MAR. Data quality varied considerably depending on the ship's speed. To correct the depth-measurements for a varying sound velocity (SV) in the water column, SV-profiles were taken regularly by using an SVPlus probe from Applied Microsystems, which directly measures the sound speed every 1 s during its deployment[35,36]. Shallow seafloor (<200 mbsl) in the vicinity of Nightingale Island was mapped using the KONGSBERG EM1002 MBES with 111 beams, mounted temporarily in the moon-pool of the vessel during the recovery of land stations[35]. Beside the depth information, the EM120/122 and EM1002 systems also provide seabed and water column backscatter data. Rather than using the per-beam backscatter data, we extracted the higher-resolution seabed image reflectivity (also referred to as "beam time series" or "pseudo side-scan" data) from the raw MBES files. To avoid confusion within this non-standardized terminology, we have opted to use the term "seabed backscatter intensity". Due to the different transducer geometry and frequency, EM1002 backscatter has not been considered for the combined mosaic.

Data were edited and processed using the open source software package MB-System[62]. Due to the sometimes high noise level of the multibeam data, we decided not to apply automatic filters and to edit the data manually using *mbeditviz*. High-resolution digital elevation models were generated with the *mbgrid* tool in 30 and 50 m grid cell resolutions.

The backscatter intensity values were corrected by calculating average amplitudes as a function of the seafloor grazing angle. A maximum swath angle of 64° was chosen and the grazing angle was calculated using the across-track slope from the bathymetry. An angular varying gain over 300 pings was applied. Finally, the backscatter data were gridded using a Gaussian weighted mean filter. In

addition, to fill grid cells not filled by swath data, a spline interpolation was applied with a radius of three times the grid cell spacing, which was set to 30 and 50 m (5 m for EM1002). Taking advantage of the different algorithms implemented in MB-System[62] and in the QPS Fledermaus™ tool FMGT (Geocoder algorithm), we processed results with both packages and finally decided to use the FMGT product for a zonal/textural interpretation and classification as used in this paper. Both results are available in the PANGAEA data repository (see below). Backscatter intensity values for volcanic cones or less sediment-draped lava flows are significantly increased compared to the overall values (see Supplementary Fig. 6). Further details about backscatter validation and mosaicking are given in the Supplemental information (Supplementary Fig. 7, Supplementary Note 1).

**Rugosity raster & Benthic Terrain Modeler**. To investigate the structural complexity of the study area and to assure a level of confidence when interpreting other derivates, a rugosity (surface roughness) raster was created using the arc–chord ratio (ACR) index[63]. The ACR method is defined as the contoured area of the surface divided by the area of the surface orthogonally projected onto a plane of best fit. In this way, it effectively decouples rugosity from the slope. To reduce artefacts in the rugosity raster caused by the underlying bathymetry data, the ACR rugosity raster was calculated from the threefold resampled bathymetry raster.

The benthic terrain of TdC was analyzed using ESRI's ArcGIS™ with the aid of the application's BTM v. 3.0[34,64] which classifies bathymetry data and analyses seafloor characteristics.

The most important derivative for this analysis is the Bathymetric Position Index (BPI). The BPI is modified from the Topographic Position Index, which is used in analyses of terrestrial environments[65]. It compares the elevation of each cell in the bathymetry raster with the mean elevation of a defined neighborhood around that cell. In this analysis, an annular neighborhood with an inner and outer radius is used. To identify both fine and broad features on the seafloor, two BPI grids were created using an inner radius of 15 km and an outer radius of 22.5 km for the broad-scale BPI raster and 1.5 and 3 km, respectively, for the fine-scale BPI raster. Since the bathymetric position tends to be autocorrelated[66], the BPI grids were standardized. The overall structural analysis was based on a classification dictionary (Supplementary Table 2) which defines several geomorphological structures by their broad and fine-scale BPI and their slope. The transition from flat areas to broad slopes was set to 3° and from broad to steep slopes to 25°. In order to compare and validate our BPI products we used the rugosity derived from bathymetry.

**Volume calculations of volcanic cones**. We utilized the GIS software Global Mapper v18.2 for volume calculations of the volcanic cones, though we generated the volcanic cone polygons with ESRI's ArcGIS™. The result of the BTM classification was used to extract the four classes that represent outcrops or local ridges (see Supplementary Table 2: Classes 6, 11, 13, and 14). This outcrop raster was then converted into polygon features. Areas smaller than 0.1 km² were excluded from further analyses. Since it is not possible to select polygons by their shape, a manual selection of the polygon features was necessary to remove all polygons that do not represent circular-based cone-shaped morphologies (e.g., elongated ridges or parts of the main islands). Furthermore, all polygons were buffered because the outcrop classes from the BTM classification do not extend to the approximate base of the outcrops, due to the functionality of the BPI. The best horizontal buffer distance was determined to be 200 m. Finally, the polygons were smoothed by a Bezier Interpolation and exported as shapefiles. The actual volume calculation was conducted with Global Mapper's analysis & measurement tool "Pile Volume" and exported as a.CSV file (Supplementary Table 1).

Mean backscatter values were calculated for each volcanic cone by using the ArcGIS tool "Zonal Statistics." Very low backscatter values (<−48 dB) were excluded beforehand using the "Extract by Attribute" tool as they mainly occur near the outer beams and the nadir and, therefore, most likely represent erroneous dB values.

**Sub-bottom profiler data**. Sub-bottom profiler measurements were conducted by means of Teledyne Parasound DS3 (P70) parametric sediment echosounder during both R/V MARIA S. MERIAN cruises until the shutdown of the system due to technical reasons on cruise MSM24. The system's beam width is 4.5° along-track and 5.0° across-track. The primary high frequency signal (18 kHz) was recorded for water-column investigations, while the secondary low frequency signal (SLF, 4 kHz) is used to image the shallow subseafloor.

The Parasound files were converted into SEG-Y data by using the PS32segy tool (Hanno Keil, University of Bremen, Germany). Afterwards, the ReflexW software[67] was used to low-pass filter the data and to mute the water column.

The vertical resolution of reflections limits the sediment layer separation and depends on the pulse length, which is defined by its frequency and number of periods. Parasound beams are characterized by 4 kHz SLF frequency and a periodic time of ¼ kHz = 0.25 ms. During MSM20/2 the continuous wave (CW) pulse had two periods with a pulse length of 0.5 ms. Taking two-way travel time (TWT) into account, the vertical resolution in meters is half the pulse length multiplied by the SV. With an estimated SV in water of 1500 m s$^{-1}$, the vertical resolution is thus 0.5 ms × 1500 m s$^{-1}$/2 = 0.375 m. Over flat and relatively hard seafloor, the range

detection (like the depth detection of the top layer with edge detection) mainly depends on the signal bandwidth of the pulse. The higher the bandwidth, the steeper the rising edge of the pulse. With CW pulses, the detection resolution is nominally half the vertical acoustic resolution. For the example above, the detection resolution then is 0.19 m. The sample rate resolution is given by the sample rate of the receiving signal, in our case 12 kHz. Considering SV for the TWT in water, the sample rate resolution results to ~6 cm ((1500 m s$^{-1}$/2)/(12,200 s$^{-1}$)).

Note: the sample rate resolution is always smaller than half the vertical acoustic resolution, which in our case is 0.19 m, as mentioned above.

**Satellite data investigation**. We tried to back-track pumice of the 2004 eruption in MODIS/Terra + Aqua satellite data using the technique of Jutzeler et al.[61]. However, the 2004 dataset does not cover all of the spectral bands necessary for that technique, and cloud cover prevented observations in the visible light bands (surface reflectance).

## Data availability
The bathymetric grids, backscatter mosaics, and its references to the raw datasets are available through the PANGAEA archive (https://www.pangaea.de/)[68]. The parent publication link https://doi.org/10.1594/PANGAEA.906154 leads to all shown data and derivates, including the GIS analyses and products, and the processed sub-bottom profiler data.

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

## Acknowledgements

This work was supported by the German Research Foundation (DFG) grant GE 1783/4-1/2 as part of the DFG Priority Program SPP1375 SAMPLE (JE 296/9-1, 9-2). Additional support was provided by the Alfred Wegener Institute Bremerhaven. We thank the captain Ralf Schmidt, the crew of R/V MARIA S. MERIAN and the Scientific Parties of cruises MSM20/2 and MSM24. Bathymetric data acquisition during cruise MSM24 was supported by Irena Schulten and Gesa Barkawitz. This work contains public sector information, licensed under the Open Government Licence v2.0, from UKHO. Julia Sigwart and Joachim Reuter also greatly contributed to the improvement of this paper.

## Author contributions

W.H.G., P.W., T.S. and A.S. carried out the interpretation. P.W., J.K., and T.S. compiled all bathymetric data. P.W. created the backscatter mosaic and analysis. P.W. and T.S. performed BTM and statistical analyses. W.H.G., M.M., M.J., and W.J., designed the experiment and collected the data. W.H.G., P.W., M.M., and G.E. wrote the paper with contributions from all co-authors.

## Funding

## Competing interests

The authors declare no competing interests.
