## [Peer Review File · Nature Communications]

Reviewers' Comments:

Reviewer #1:

Remarks to the Author:

This is a very welcome and detailed description of the bathymetry and geology of the submarine area around the volcanic island Tristan da Cunha in the Atlantic. The new data provides this kind of information for the first time from this area. Tristan da Cunha is a classic example of an ocean island on a 'hot-spot' trail, so I would consider the paper to be of strong international interest. The features are well-described, using a very good data set. There is a considerable amount of interpretation, and mapping tools, like the benthic terrane model, are used to describe the sea floor morphology in terms of its volcanology and sedimentation. I think that this is a good use of these tools, and something that I haven't seen done in this way before to describe seafloor volcanology.

The interesting conclusion is that there has been a large amount of fairly recent volcanism on the seafloor in the area around Tristan da Cunha. This is explored in terms of seamount evolution and hotspot track models. This interpretation seems to be correct to me, and shows that the area has exciting possibilities for looking at widespread recent intraplate volcanism on the seafloor at a youthful stage in terms of building volcanic seamounts. It also has implications for hot-spot and mantle melting models for Walvis Ridge, its hot-spot track and the interaction with the mid-Atlantic Ridge.

Minor points

Data sources and coverage: Lines 84 and figs 2 and 3. The data sources for the bathymetry data are not completely clear. Line says that 'a complete coverage exists except for the direct vicinity of the islands'. Looking at fig 2 it seems as if there is a gap in data around and between the islands. But in Fig 3, this gap seems to disappear, and there is good data around the islands except for close to the shore lines. I'm sure there is a good explanation for where all the data comes from. Why not just say that a complete, merged data set was produced using all sources, and then present just this data set?

Line 74. instrument should be instruments.

Line 106. Replace coarser with coarse.

Line 107. It is not clear what the 'here' refers to.

Line 108. Insert 'with' before 'intermixed'.

Lines 109-111. I can't see the 'more coarse material including rafted blocks that can be attributed to landslides' to the NW of Tristan. I looked at Holcombe and Searle, and the new bathymetry does not seem to back up their idea that there is a large landslide NW of Tristan. NW Tristan seems to have a generally smooth, conical form, with no major collapse, but with several gullies or chutes indicating probably many small volume debris flow events. There is a hint of sediment wave structures associated with the gullies. It looks to me that Holcombe and Searle got it wrong.

Line 116. SE should be SW, I think.

Line 134. Should 'islands be island'?

Line 135. 'From about 160 to 170'. Shoaling means shallowing, so it can't shoal from 160 to 170m.

Line 136. What is a failed flank?

Line 143. Replace 'the survey allowed to map' with 'there are'.

Line 145. Replace 'we identified two areas with' with 'areas A and B have'.

Line 186. Deplete 'without having...data'.

Line 244. Replace 'cause' by 'causes'.

Line 256. Replace 'evolved' by 'formed'.

Line 262. Replace 'spans across the' with 'images'.

Line 274. Delete 'anymore'.

Lines 284-285. 'and in a view...zone'. It reads as if you can't make up your mind. Is there a mistake in the sentence. Or delete these words.

Line 285. Delete 'for'.

Line 321. Replace 'concur' with 'concur'. Actually, concurs is probably not a good word here. Maybe put: 'is in the same area as' instead.

Lines 361-362. I don't think you are clear in your thinking or are not expressing it well. Are you trying to explain why there are no eruptions within the fracture zone, or south of it? Although the FZ probably has

thin crust, is there any reason to think the FZ has thin lithosphere?. It originated as a cold zone, and may have developed thicker lithosphere. I'm not sure what seismics says. I don't see why magmas would flow in dykes laterally along the FZ either in the crust or mantle. So in summary, I don't think there is a reason to think that FZ would be prone to later focusing of magmas.

Line 368. Replace 'also' with 'the'. Replace 'decrease' with 'decreases'.

Line 370. Move 'mapped' to before 'volcanic'.

Line 373. Delete 'already'.

Line 384. Replace 'taken' by 'taking'.

Line 392. I would prefer the section heading not to be a question. I would suggest; 'Location of the 2004 submarine eruption'.

Line 413-415. I understand what you are saying, but the sentence seems clumsy. Maybe put. 'This is shallower than the 500 m depth limit for producing pumice in submarine eruptions suggested by Allen et al'.

Lines 424-427. This does not add anything, and I suggest delete.

Fig. 3. The contour interval is not clear. There is a 3500m contour, then a 3000m one, then an unlabelled contour, then a 1000m one. I expect the contour labels were automatically generated in Arc, which is not a very good way of doing it. Its best to switch off the contour labelling tool, and then label by hand.

Reviewer #2:

Remarks to the Author:

What are the major claims of the paper? Geissler et al present new seafloor bathymetry mapping data around Tristan de Cunha island and interpret the observed pattern of volcanic edifices in terms of possible hotspot activity. Estimates of the volume of erupted material are made and compared with some island and larger seamount observations in the same area. The evident recent-ness of the seafloor volcanics is combined with prior deep geophysical imaging and seismicity results to support interpretation that the current site of Walvis Ridge hotspot is located west of Tristan de Cunha, not below the island itself.

The hypothesis cannot be proven until lava sample geochemistry and age dates are obtained, but the new data are an important piece of the puzzle- if no (or very few) recent volcanics had been observed, one would infer that the plume is not active west of the island or its melt supply is very limited / unable to develop sufficient pressure to erupt.

Are they novel and will they be of interest to others in the community and the wider field? Hotspot distribution, evolution, and activity is of significant interest to geoscientists and, conceptually, to a wider scientific audience. Walvis Ridge is a key hotspot expression whose source motion & melting relate to fundamental planetary evolution. So, yes, I expect new information about this hotspot will be of interest. Having said this, I think the manuscript needs to include 1-2 new paragraphs that actually explain this importance. Also, a clearer statement about what the new seafloor mapping brings to the topic is needed- the text covers the description of observations and some comparison to other hotspots quite well, but it never comes right out to state the implication (to date) of their findings.

More specifically, the Introduction would benefit from a first paragraph that explains why it is interesting/important to know the actual location and level of activity for the current hot spot. As currently written, the paper doesn't 'show off' what is exciting about the (well explained) new mapping results and why having initial groundtruth on prior seismicity/imaging interpretations is important (i.e. indication that seafloor eruption may have occurred in 2004 and that it probably has gone on for a while (high reflectivity).

do you feel that the paper will influence thinking in the field?

These findings will likely form an important part of a strong case to return to the area and obtain a systematic suite of samples from the recent volcanics so that geochemical and age-dating analyses can be done to test the hypothesis that Walvis Ridge hotspot extends past Tristan de Cunha, the previously inferred

current site.

comment on the appropriateness and validity of any statistical analysis, as well the ability of a researcher to reproduce the work

Table S1 needs to include identifier/lat-lon for each cone. In PDF version, only the volume/area are listed with no way to relate those measures to a feature on the map.

Additional comments:

The title employs 'hydro-acoustic' in a way that is not typical, which may be confusing to readers, even if it is not strictly incorrect. The new evidence presented in the paper is bathymetry, almost all from shipboard multi-beam sonar systems, and backscatter data. In the marine geoscience (and nuclear test monitoring) community, hydro-acoustic typically refers to signals that travel a significant lateral distance in the water column, and almost always the source is not co-located with the receiver. Using 'sonar' or 'seafloor swath mapping' might be more clear.

Additional sentence(s) that discuss the estimated volume of newly-mapped seafloor volcanics in terms of what this might mean in for the level of hotspot magmatism (recently, in light of what's know for Walvis activity over time, perhaps) would help clarify the intent of that part of the study.

The interpretation of the OCC in terms of reduced plume-ridge interaction is one viable scenario. Another might be that magma supply is more than typically episodic, which could suggest pulsing of plume-enhanced melting. Authors might want to consider this and mention it if they agree or have evidence one way or another. The last sentence of the Abstract may warrant some revision, if so.

The grammar/phrasing is awkward in a number of places throughout the text. I have suggested ways to improve this in that annotated PDF.

Donna Blackman

Reviewer #3:

Remarks to the Author:

Summary: The authors present multibeam and backscatter and shallow sub-bottom data to propose the juvenescence of the Walvis Ridge towards the west, near the Tristan de Cunha Island hot spot track. They identify possible monogenetic submarine volcanoes in this vicinity and identify the Isolde Seamount as a possible eruptive formation from the 2004 event.

The paper presents new remote sensing (multibeam and sub-bottom data) and describe the details that lead to their conclusion of the Isolde Seamount as likely active in 2004. They also describe the short-comings due to a lack of ground-truthing (rock sample data). The interpreted multibeam maps and data from the sub-bottom profiler provide evidence to support their hypothesis. I believe this paper presents important new information, which can be further verified with physical samples.

Specific Comments:

Line 69: colloquialism: change "till" to "until"

Line 97 & Line 165: Use parenthetical citation of figure rather than calling it out in the text as the subject of the sentence

Line 114: in several places English needs to be cleaned up. This sentence is one example "Similar to Tristan da Cunha..." would be better grammar than "As for"

Line 379: "proof or disproof" should be "prove or disprove"

Figure 2: Inset map to show location isn't zoomed out enough. Need to show continents to make this clearer. It would also be nice for this map to include letters or something that corresponds to the other figures that follow so it can more easily be seen where each fits in.

Figure 7: Not sure this is necessary to the paper.

Nicole Raineault

Reviewer #4:

Remarks to the Author:

The manuscript entitled "The juvenescence of the Walvis ridge – evidence for recent submarine volcanism from hydro acoustic data in the vicinity of Tristan de Cunha" presents a suite of recently collected and high-quality (although incomplete) geophysical data across the westernmost region of the Walvis Ridge, west and southwest of Tristan de Cunha. These geophysical data nicely show numerous seafloor features including abyssal hill fabric, seafloor volcanism and core complexes.

The results of multibeam data analysis suggest there is a volcanic province made up of small and potentially young monogenetic cones. These data enable the authors to calculate morphometric parameters of the main island of Tristan de Cunha, the nearby volcanic province and the dominant tectonic fabric. The acoustic backscatter is presented in a non-standard colour bar which is confusing, I would advise the authors to opt for the standard grey-scale colour bar for consistency.

Sub-bottom profiler data are used to interrogate the shallow subsurface showing regions that are sedimented or regions of low acoustic penetration (deduced to be hard substrate likely volcanic), and relative age relationships between these substrates. Age relationships between the volcanics and sediments are inferred from these data, suggesting the volcanics are younger than the sedimentary sequence. Unfortunately, due to the size and vertical exaggeration in Figure 8, it is difficult to identify the structures discussed in text.

The results section of this manuscript presents a descriptive analysis of multibeam bathymetry and sub bottom profiler data that support previous studies of an age progressive seamount chain for the Walvis Ridge. The discussion section draws on other global examples of intraplate volcanism and relevant literature. Given these data are purely geophysical, there is a large amount of speculation surrounding the stages origins and age of volcanism. There are some discussion points interwoven within the results section (e.g., line 173/4 – "...indicating younger overprint by volcanic activities" and line 107/8 – "...indicating finer erosional material intermixed pelagic sedimentation"). I suggest keeping the results and interpretation separated throughout.

Most of the discussion points, whilst interesting, don't specifically discuss the analyses of the data that are presented in the study (e.g., the morphometric analysis of seamounts/cones). There is no solid concluding statement or paragraph explaining how the presented dataset provides new evidence and/or significantly progresses research in the region or how the new data progress the fields of hotspot volcanism and tectonics. This needs to be seriously reconsidered.

The language in the manuscript could be improved and/or formalised throughout (e.g. "We also tried to track the pumice by satellite data..." (Line 424- quite informal language) and "we cannot proof or disproof any recent interaction..." (line 379 - spelling errors). In my opinion the title requires revision for 2 reasons: firstly, it overstates what the manuscript presents – the age progression of the Walvis Ridge is not presented in this study there are no age dates in this study, and this finding has been presented in previous studies. This study presents a geomorphic evaluation of a volcanic province, an oceanic core complex and tectonic fabric near Tristan de Cunha. Secondly, the word "juvenescence" is confusing and unnecessary. I am a native English speaker and I found this word over-complicated the sentence, and it could serve to potentially alienate other readers.

Finally, although I believe it is worthwhile that these data and analyses are published, I question the whether this manuscript adequately fulfils the requirements of Nature Communications. According to the website (https://www.nature.com/authors/policies/peer_review.html), the criteria for publication in Nature Research journals are studies that:

1. Provide strong evidence for its conclusions.

o The current study does not provide strong evidence for an age progressive seamount chain as there is no actual age dates presented. Relative ages are speculated.

2. Are novel (we do not consider meeting report abstracts and preprints on community servers to compromise novelty).

o These findings do not appear to be particularly novel as there are published studies with $^{40}\text{Ar}/^{39}\text{Ar}$ dates showing age progression along the Walvis Ridge (Rohde et al., 2013). Speculating the continuation of the seamount chain to the west of Tristan de Cuhna is dangerous without the aid of rock samples to compare magma composition and ages of the volcanic province. Particularly as the authors clearly state in the introduction that there is "an ongoing debate, if the volcanism at Tristan de Cuhna is either caused by an underlying mantle plume or by plate tectonics and shallow mantle convection" (line 57-60).

3. Are of extreme importance to scientists in the specific field.

o These data are interesting, high-quality and important to publish; however this study does not represent an advance in understanding likely to influence thinking in the field of hotspot volcanism and/or seamount chain formation.

4. Ideally, interesting to researchers in other related disciplines.

o These data may be of interest to other disciplines, however the relevance of these data to other disciplines is not explored in this manuscript.

Based on these criteria, I suggest the overall pitch of this manuscript should be refocused to more adequately represent the results and analyses presented, and resubmitted to a more specialised journal.

Dear Reviewers,

thank you very much for the valuable comments and suggestions. We now incorporated most of them in the revised version of the manuscript. Our response is marked with “→”.

With best regards
Wolfram Geissler

Reviewers' comments:

Reviewer #1 (Remarks to the Author):

This is a very welcome and detailed description of the bathymetry and geology of the submarine area around the volcanic island Tristan da Cunha in the Atlantic. The new data provides this kind of information for the first time from this area. Tristan da Cunha is a classic example of an ocean island on a ‘hot-spot’ trail, so I would consider the paper to be of strong international interest. The features are well-described, using a very good data set. There is a considerable amount of interpretation, and mapping tools, *like the benthic terrane model, are used to describe the sea floor morphology in terms of its volcanology and sedimentation.* I think that this is a good use of these tools, and something that I haven’t seen done in this way before to describe seafloor volcanology.

The interesting conclusion is that there has been a large amount of fairly recent volcanism on the seafloor in the area around Tristan da Cunha. This is explored in terms of seamount evolution and hotspot track models. This interpretation seems to be correct to me, and shows that the area has exciting possibilities for looking at widespread recent intraplate volcanism on the seafloor at a youthful stage in terms of building volcanic seamounts. It also has implications for hot-spot and mantle melting models for Walvis Ridge, its hot-spot track and the interaction with the mid-Atlantic Ridge.

Minor points

Data sources and coverage: Lines 84 and figs 2 and 3. The data sources for the bathymetry data are not completely clear. Line says that ‘*a complete coverage exists except for the direct vicinity of the islands*’. Looking at fig 2 it seems as if there is a gap in data around and between the islands. But in Fig 3, this gap seems to disappear, and there is good data around the islands except for close to the shore lines. I’m sure there is a good explanation for where all the data comes from. Why not just say that a complete, merged data set was produced using all sources, and then present just this data set?

→ This part of the manuscript was revised to clarify these points.

Line 74. instrument should be instruments.

Line 106. Replace coarser with coarse.

Line 107. It is not clear what the ‘here’ refers to.

Line 108. Insert ‘with’ before ‘intermixed’.

Lines 109-111. I can’t see the ‘more coarse material including rafted blocks that can be attributed to landslides’ to the NW of Tristan. *I looked at Holcombe and Searle, and the new bathymetry does not seem to back up their idea that there is a large landslide NW of Tristan.* NW Tristan seems to have a generally smooth, conical form, with no major collapse, but with several gullies or chutes indicating probably many small volume debris flow events. There is

a hint of sediment wave structures associated with the gullies. It looks to me that Holcombe and Searle got it wrong.

→ We have extended a little in that direction and included an interpretation in Figure 3d, that shows lobes of slope failure deposits.

Line 116. SE should be SW, I think. → No, SE is right.

Line 134. Should 'islands be island? → Both have "Island" in the right name. Revised.

Line 135. 'From about 160 to 170'. Shoaling means shallowing, so it can't shoal from 160 to 170m.

Line 136. What is a failed flank? → Revised.

Line 143. Replace 'the survey allowed to map' with 'there are'.

Line 145. Replace 'we identified two areas with' with 'areas A and B have'.

Line 186. Deplete 'without having...data'.

Line 244. Replace 'cause' by 'causes'.

Line 256. Replace 'evolved' by 'formed'.

Line 262. Replace 'spans across the' with 'images'.

Line 274. Delete 'anymore'.

Lines 284-285. 'and in a view...zone'. It reads as if you can't make up your mind. Is there a mistake in the sentence. Or delete these words. → Revised.

Line 285. Delete 'for'.

Line 321. Replace 'concur' with 'concur'. Actually, concurs is probably not a good word here. Maybe put: 'is in the same area as' instead.

Lines 361-362. I don't think you are clear in your thinking or are not expressing it well. Are you trying to explain why there are no eruptions within the fracture zone, or south of it? Although the FZ probably has thin crust, is there any reason to think the FZ has thin lithosphere?. It originated as a cold zone, and may have developed thicker lithosphere. I'm not sure what seismics says. I don't see why magmas would flow in dykes laterally along the FZ either in the crust or mantle. So in summary, I don't think there is a reason to think that FZ would be prone to later focusing of magmas.

→ This part was revised for clarification.

Line 368. Replace 'also' with 'the'. Replace 'decrease' with 'decreases'.

Line 370. Move 'mapped' to before 'volcanic'.

Line 373. Delete 'already'.

Line 384. Replace 'taken' by 'taking'.

Line 392. I would prefer the section heading not to be a question. I would suggest; 'Location of the 2004 submarine eruption'.

→ Revised.

Line 413-415. I understand what you are saying, but the sentence seems clumsy. Maybe put. 'This is shallower than the 500 m depth limit for producing pumice in submarine eruptions suggested by Allen et al'.

→ Revised.

Lines 424-427. This does not add anything, and I suggest deplete.

Fig. 3. The contour interval is not clear. There is a 3500m contour, then a 3000m one, then an unlabelled contour, then a 1000m one. I expect the contour labels were automatically

generated in Arc, which is not a very good way of doing it. Its best to switch off the contour labelling tool, and then label by hand.

→ More contour labels were added.

All other minor comments and suggestions were incorporated into the revised version.

Reviewer #2 (Remarks to the Author):

What are the major claims of the paper? Geissler et al present new seafloor bathymetry mapping data around Tristan de Cunha island and interpret the observed pattern of volcanic edifices in terms of possible hotspot activity. Estimates of the volume of erupted material are made and compared with some island and larger seamount observations in the same area. **The evident recentness of the seafloor volcanics is combined with prior deep geophysical imaging and seismicity results to support interpretation that the current site of Walvis Ridge hotspot is located west of Tristan de Cunha, not below the island itself.**

The hypothesis cannot be proven until lava sample geochemistry and age dates are obtained, but the **new data are an important piece of the puzzle- if no (or very few) recent volcanics had been observed, one would infer that the plume is not active west of the island or its melt supply is very limited / unable to develop sufficient pressure to erupt.**

Are they novel and will they be of interest to others in the community and the wider field?

Hotspot distribution, evolution, and activity is of significant interest to geoscientists and, conceptually, to a wider scientific audience. Walvis Ridge is a key hotspot expression whose source motion & melting relate to fundamental planetary evolution. So, yes, I expect new information about this hotspot will be of interest. **Having said this, I think the manuscript needs to include 1-2 new paragraphs that actually explain this importance.** Also, a clearer statement about what the new seafloor mapping brings to the topic is needed- the text covers the description of observations and some comparison to other hotspots quite well, **but it never comes right out to state the implication (to date) of their findings.**

More specifically, **the Introduction would benefit from a first paragraph that explains why it is interesting/important to know the actual location and level of activity for the current hot spot.** As currently written, the paper doesn't 'show off' what is exciting about the (well explained) new mapping results and why having initial groundtruth on prior seismicity/imaging interpretations is important (i.e. indication that seafloor eruption may have occurred in 2004 and that it probably has gone on for a while (high reflectivity)).

→ The introduction has undergone a complete revision to document the importance of the presented data.

do you feel that the paper will influence thinking in the field?

These findings will likely form an important part of a strong case to return to the area and obtain a systematic suite of samples from the recent volcanics so that geochemical and age-dating analyses can be done to test the hypothesis that Walvis Ridge hotspot extends past Tristan de Cunha, the previously inferred current site.

comment on the appropriateness and validity of any statistical analysis, as well the ability of a researcher to reproduce the work

Table S1 needs to include identifier/lat-lon for each cone. In PDF version, only the volume/area are listed with no way to relate those measures to a feature on the map.

→ Revised.

Additional comments:

The title employs 'hydro-acoustic' in a way that is not typical, which may be confusing to readers, even if it is not strictly incorrect. The new evidence presented in the paper is bathymetry, almost all from shipboard multi-beam sonar systems, and backscatter data. In the marine geoscience (and nuclear test monitoring) community, hydro-acoustic typically refers to signals that travel a significant lateral distance in the water column, and almost always the source is not co-located with the receiver. Using 'sonar' or 'seafloor swath mapping' might be more clear.

Additional sentence(s) that discuss the estimated volume of newly-mapped seafloor volcanics in terms of what this might mean in for the level of hotspot magmatism (recently, in light of what's know for Walvis activity over time, perhaps) would help clarify the intent of that part of the study.

The interpretation of the OCC in terms of reduced plume-ridge interaction is one viable scenario. Another might be that magma supply is more than typically episodic, which could suggest pulsing of plume-enhanced melting. Authors might want to consider this and mention it if they agree or have evidence one way or another. The last sentence of the Abstract may warrant some revision, if so.

→ The parts related to OCC are completely rewritten.

The grammar/phrasing is awkward in a number of places throughout the text. I have suggested ways to improve this in that annotated PDF.

→ Thank you very much for all the suggestions that were incorporated in the revised version. Grammar/phrasing was now also checked by a native speaker again.

Donna Blackman

Reviewer #3 (Remarks to the Author):

Summary: The authors present multibeam and backscatter and shallow sub-bottom data to propose the juvenescence of the Walvis Ridge towards the west, near the Tristan de Cunha Island hot spot track. They identify possible monogenetic submarine volcanoes in this vicinity and identify the Isolde Seamount as a possible eruptive formation from the 2004 event.

The paper presents new remote sensing (multibeam and sub-bottom data) and describe the details that lead to their conclusion of the Isolde Seamount as likely active in 2004. They also describe the short-comings due to a lack of ground-truthing (rock sample data). The interpreted multibeam maps and data from the sub-bottom profiler provide evidence to support their hypothesis. I believe this paper presents important new information, which can

be further verified with physical samples.

Specific Comments:

Line 69: colloquialism: change “till” to “until”

Line 97 & Line 165: Use parenthetical citation of figure rather than calling it out in the text as the subject of the sentence

→ We reduced the use of “Figure” as the subject of the sentence.

Line 114: in several places English needs to be cleaned up. This sentence is one example “Similar to Tristan da Cunha...” would be better grammar than “As for”

→ Grammar/phrasing was now also checked by a native speaker again.

Line 379: “proof or disproof” should be “prove or disprove”

Figure 2: Inset map to show location isn't zoomed out enough. Need to show continents to make this clearer. It would also be nice for this map to include letters or something that corresponds to the other figures that follow so it can more easily be seen where each fits in.

→ We included the references to other figures. However, we think the inset map is OK, showing the major features around the study area. The relation to the southern part of Africa is already shown in figure 1.

Figure 7: Not sure this is necessary to the paper.

→ We agree that this figure is not that important for the main manuscript. We now include it in the supplementary material for documentation.

Nicole Raineault

Reviewer #4 (Remarks to the Author):

The manuscript entitled “The juvenescence of the Walvis ridge – evidence for recent submarine volcanism from hydro acoustic data in the vicinity of Tristan de Cunha” presents a suite of recently collected and high-quality (although incomplete) geophysical data across the westernmost region of the Walvis Ridge, west and southwest of Tristan de Cunha. These geophysical data nicely show numerous seafloor features including abyssal hill fabric, seafloor volcanism and core complexes.

The results of multibeam data analysis suggest there is a volcanic province made up of small and potentially young monogenetic cones. These data enable the authors to calculate morphometric parameters of the main island of Tristan de Cunha, the nearby volcanic province and the dominant tectonic fabric. **The acoustic backscatter is presented in a non-standard colour bar which is confusing, I would advise the authors to opt for the standard grey-scale colour bar for consistency.**

→ We included a version with grey-scale color in the Supplementary Materials now for documentation. For the purpose of a classification, we stick with our colors to clearly distinguish between the classes.

Sub-bottom profiler data are used to interrogate the shallow subsurface showing regions that are sedimented or regions of low acoustic penetration (deduced to be hard substrate likely volcanic), and relative age relationships between these substrates. **Age relationships between the volcanics and sediments are inferred from these data, suggesting the volcanics are younger than the sedimentary sequence. Unfortunately, due to the size and vertical exaggeration in Figure 8, it is difficult to identify the structures discussed in text.**

→ Original Figure 8 (now Figure 7) was revised to better highlight the details.

The results section of this manuscript presents a descriptive analysis of multibeam bathymetry and sub bottom profiler data that support previous studies of an age progressive seamount chain for the Walvis Ridge. The discussion section draws on other global examples of intraplate volcanism and relevant literature. Given these data are purely geophysical, there is a large amount of speculation surrounding the stages origins and age of volcanism. **There are some discussion points interwoven within the results section** (e.g., line 173/4 – “...indicating younger overprint by volcanic activities” and line 107/8 – “...indicating finer erosional material intermixed pelagic sedimentation”). **I suggest keeping the results and interpretation separated throughout.**

→ Most of the discussion is within its separate section. However, first basic interpretations have to be made already in the results section.

Most of the discussion points, whilst interesting, don't specifically discuss the analyses of the data that are presented in the study (e.g., the morphometric analysis of seamounts/cones). **This needs to be seriously reconsidered.**

→ The manuscript aims to support the interpretation of the current hot spot regime west of TdC. Therefore, we utilized the GIS based analyses of bathymetry and hydroacoustic backscatter next to seismological information. To refer to the complaint above, we rewrote several passages/sections also with respect to better highlight the remote sensing part of the paper.

The language in the manuscript could be improved and/or formalised throughout (e.g. “We also tried to track the pumice by satellite data...” (Line 424- quite informal language) and “we cannot proof or disproof any recent interaction...” (line 379 - spelling errors). In my opinion the title requires revision for 2 reasons: firstly, it overstates what the manuscript presents – **the age progression of the Walvis Ridge is not presented in this study there are no age dates in this study, and this finding has been presented in previous studies.** This study presents a geomorphic evaluation of a volcanic province, an oceanic core complex and tectonic fabric near Tristan de Cuhna. Secondly, the word “**juvenescence**” is confusing and unnecessary. I am a native English speaker and I found this word over-complicated the sentence, and it could serve to potentially alienate other readers.

→ Title was revised. Even if we present no new absolute ages, we discuss relative ages that allow the conclusion that the volcanic edifices west of TdC got emplaced fairly recent and could be the youngest expression of mantle plume volcanism. We deleted the word “**juvenescence**” to avoid unnecessary confusion.

Finally, although I believe it is worthwhile that these data and analyses are published, I question the whether this manuscript adequately fulfils the requirements of Nature

Communications. According to the website (https://www.nature.com/authors/policies/peer_review.html), the criteria for publication in Nature Research journals are studies that:

1. Provide strong evidence for its conclusions.
 - o **The current study does not provide strong evidence for an age progressive seamount chain as there is no actual age dates presented. Relative ages are speculated.**
2. Are novel (we do not consider meeting report abstracts and preprints on community servers to compromise novelty).
 - o These findings do not appear to be particularly novel as there are published studies with $^{40}\text{Ar}/^{39}\text{Ar}$ dates showing age progression along the Walvis Ridge (Rohde et al., 2013). Speculating the continuation of the seamount chain to the west of Tristan de Cuhna is dangerous without the aid of rock samples to compare magma composition and ages of the volcanic province. Particularly as the authors clearly state in the introduction that there is “an ongoing debate, if the volcanism at Tristan de Cuhna is either caused by an underlying mantle plume or by plate tectonics and shallow mantle convection” (line 57-60).
3. Are of extreme importance to scientists in the specific field.
 - o These data are interesting, high-quality and important to publish; however this study does not represent an advance in understanding likely to influence thinking in the field of hotspot volcanism and/or seamount chain formation.
4. Ideally, interesting to researchers in other related disciplines.
 - o These data may be of interest to other disciplines, however the relevance of these data to other disciplines is not explored in this manuscript.

Based on these criteria, I suggest the overall pitch of this manuscript should be refocused to more adequately represent the results and analyses presented, and resubmitted to a more specialised journal.

Reviewers' comments:

Reviewer #3 (Remarks to the Author):

The authors have made improvements to the manuscript that greatly improved the flow and clarity of the research narrative and its claims. I believe the paper is now suitable for publication.

Reviewer #4 (Remarks to the Author):

As mentioned in previous reviews, the manuscript that is now titled "Young Pre-Shield Volcanism above the Tristan da Cunha Mantle Plume" presents a suite of recently collected and high-quality geophysical data across the westernmost region of the Walvis Ridge west and southwest of Tristan de Cuhna. These geophysical data nicely show numerous seafloor features including abyssal hill fabric, seafloor volcanism and core complexes.

After reviewing the revised manuscript and reading the response to reviewers I still feel that this manuscript requires some additional revisions. I have summarised 4 main issues with this revised manuscript, providing examples of each issue. These issues include: 1) the literature as recounted in this manuscript, 2) the use and interpretation of acoustic seafloor backscatter, 3) speculations made in the discussion, and 4) writing style. I have then detailed more minor issues with the manuscript below.

Major issues:

1. Literature reviewed

In Line 61 – 65, the authors present a very brief literature review of the volcanism has a few substantial errors. Basanite does not equal trachybasalt these are different rock types. This is what is implied with "volcanic rocks on the island are dominated by basanites (trachybasalts)...".

The summary of the volcanology according to Hicks et al. 2012 is geologically inaccurate. I recommend going back and thoroughly reviewing the original works and that have described the volcanology of this island (which the authors have cited but incorrectly summarised).

The interpretation made by the authors that this is a shield volcano is incorrect, this is not a shield volcano. Hicks et al. 2012 do not describe this a shield volcano, this is a misinterpretation of the literature. This is an oceanic stratovolcano. This literature summary of the volcanism is an extremely important section considering the title of the paper and so should be done thoroughly and so it is geologically correct. I suggest this section is deleted and completely rewritten. Perhaps a greater emphasis on the geology literature surrounding TdC would help. It is troubling to me that this has not been clearly achieved within the introduction. Such an erroneous review of the literature so early on does not set a high bar for the rest manuscript.

2. Backscatter presentation and interpretation

I have a general issue with the way acoustic seafloor backscatter data is presented and discussed in this manuscript:

2.1 Presentation of backscatter

As I've mentioned before, the colour scale is non-standard. I suggested this to be changed in the previous review and it wasn't "for purposes of classification". I don't know where these classifications are, do you mean the figure that has a graduated colour scale? I reiterate that backscatter should be presented in standard colours so people who use it frequently are familiar.

When referring to backscatter is it clearer to say "high/low acoustic seafloor backscatter intensity" rather than high/low amplitude backscatter. It's important to emphasise that this is seafloor backscatter too, as there is more than one type of backscatter data when using EM122 data (e.g. water column backscatter). For clarity and consistency with other literature, I think it's important to change the way you refer to backscatter.

The authors have used exact decibel values in this manuscript, whilst I understand you can compare decibel

values within one voyage, it is not valid to compare absolute backscatter values between voyages. Seeing as it appears from the methods section that multiple MBES were used on multiple voyages, and there is no information about backscatter calibration then comparing dB values is not valid in this instance. Using absolute dB values is a bit misleading in any case. I believe it is more accurate to describe backscatter as relatively high/low intensity, I suggest removing any references to absolute acoustic backscatter values in decibels.

2.2 Interpretation of backscatter

The way that backscatter is overinterpreted in this manuscript is very concerning to me. For example:

Line 116-118: "High amplitude backscatter, as shown in Fig. 3b, indicates the presence of coarse material, possibly volcanic sediments"

Line 119-121: "Low backscatter amplitudes indicate that the aprons consist of generally fine-grained volcanoclastic intermixed or draped with pelagic or hemipelagic sediments"

There are no samples presented in this study so the readers can only assume you are interpreting geology based purely on backscatter intensity. This is highly problematic. There are examples where backscatter has been correlated with seafloor substrates, however ONLY where adequate ground truth data is available. Seafloor backscatter without ground truth and without proper calibration and processing, provides extra information about the hardness of the seafloor and can be used to INFER substrate. High backscatter does not indicate the presence of coarse material, similarly, low backscatter does not always indicate fine-grained material and does not allow you to infer sediment composition. From the descriptions of the seafloor geology it seems like there are geologic samples, but they are not presented?

I suggest the authors re-evaluate of the way that backscatter is presented to be more consistent with other papers and most importantly revise the interpretation of the backscatter data to reflect what these data can accurately tell us. Alternatively present the geology that backs up your geophysical data.

3. Speculation and discussion

There is a lot of speculation and the discussion does not feed back into the results presented. As a reader, when I get to the discussion, I want to know what the data presented tells us (that we didn't know before) about the volcanism in the region. I don't feel this has been achieved.

The data as they are presented in this study, can tell us about the surface and shallow subsurface geomorphology of the region. Not the age and not the rock/sediment composition. I think that it is important not to overstate what the data is telling us.

Discussing the ages of volcanism as being "young" because they "look young" is fraught with complications. It is a dangerous speculation since time and time again Argon dating is showing that all age progressions indicated by edifice shape/preservation are not in fact real or not as linear as they were speculated to be. A hard seafloor alone, without any supporting evidence for age does not automatically equal young volcanism in my opinion.

4. Writing style

The writing style needs work. It is quite informal, repetitive and in places very hard to follow due to the long sentences.

For example: Line 199 – 202: "Evidence for volcanic activity in the form of circular feature and W-E oriented ridges seems to extend out to 250 km westwards of the TdC island group to about 14 48'W, where we mapped a small volcanic cone on top of old abyssal hills (~10 Myr old ocean crust)."

Also, there are a few sentences that don't seem to add much value – the authors should be more concise with writing, particularly for a high impact journal such as Nature. If it's not relevant, perhaps leave it out. e.g., line 99-101: "Further away from the islands and the MAR, bathymetry data were collected along the cruise tracks in-between seismological and electromagnetic ocean-bottom stations and on transit to the area36,37" – to me this sentence adds nothing.

Line 402 "although this region is less well covered by bathymetry data" – the wording here could be revised to be clearer.

Repetitive use of "the high-resolution data set" to begin sentences could be changed to make it easier to read.

Further to this, the use of acronyms in this manuscript results in a highly convoluted confusing read. e.g., Line 417 – "The absence of an OCC at the inside corner of the northern RTI on the TdCTFFZS suggests melt supply there may be slightly enhanced by the presence of the Tristan mantle plume." Acronyms are useful when trying to avoid long confusing sentences with repetitive concepts. The use of acronyms here isolates and confuses the reader, making this much harder work to read. It is in the best interests of the authors to write a clear and concise manuscript, so people want to read (and cite) it.

Minor issues:

The title capitalises Pre-Shield, Volcanism, Mantle and Plume. This isn't necessary. The title should be "Young pre-shield volcanism above the Tristan da Cunha mantle plume"

Line 20: change potential to potentially

Line 22: "with multibeam and sediment echosoundings" – I think it is more correct (and more common) to say multibeam echo sounds (MBES) and sub-bottom profiler. Its good to call these tools the same as what other people call them so everyone is on the same page.

Line 38: "The islands are the emergent parts of well-developed shield, post-shield, or even rejuvenated volcanic edifices¹. However, little is known about embryonic stages of ocean-island growth, which begin with the very first eruptions of hot spot-related volcanic rocks onto "pristine" oceanic crust." – this doesn't make sense to me, "However" is a conjunction that should be used to convey a change from the last topic. to make this "however" necessary you need to say something like "The islands are the emergent parts of well-developed shield, post-shield, or even rejuvenated volcanic edifices that occur in the latter stages of ocean island growth. However, little is known about embryonic stages of ocean-island growth, which begin with the very first eruptions of hot spot-related volcanic rocks onto "pristine" oceanic crust."

Or get rid of the however.

Line 49: change to " ...melting relate to the fundamentals of mantle plume dynamics"

Line 118: what rise are the authors referring to here?

Line 127-128: "The lobe within the canyon between the TdC and Inaccessible Island is probably the most recently formed" – it is not clear to me what this interpretation is based on. If its only backscatter, then I think it needs to be rethought.

Line 133: 25 x 40 km = 200 km²? Where is the 200 km coming from? 25 x 40 is closer to 1000 km². Am I missing something here?

Features in the text must be clearly labelled on the figures. There are several locations where the authors are referring to a feature which is not labelled on any figure. These should be labelled to make it easier to follow, some examples that are not labelled include:

Line 161 – the SE flank that has been removed

Line 164 – the volcanic ligament and complex

Line 234 – spreading compartment corridor

Line 254 – normal abyssal hill fabric

Line 316 – faults observed in the sub bottom profiler data

Also important to specifically refer to figures for example Figs 1a and 2b not just Figs 1 & 2

The volcanic structures described in 177-180: have been interpreted to follow a specific trend, which is fine. I feel as though this could be easily backed up with more robust statistical information rather than just having a qualitative comment about how they "seem" to trend. Again, in line 405: "it seems that the strike of the abyssal hills...". This isn't very scientific.

Line 218: change 3.0 km to 3 km, and the 1.0 – 2.0 km to 1-2 km. the decimal place adds nothing,
Line 261: I think the authors need to be more specific about the source of the earthquake epicentres. At the very least spell out the acronym NEIC and provide a website or location of the data.

The differences in satellite derived bathymetry and multibeam bathymetry described in Lines 272 – 277 are problematic. Bathymetry is predicted with the aid of both satellite altimetry measurements and soundings along ship tracks (e.g., Smith & Sandwell). Although the gravity field is used to predict bathymetry, using the dominating effect of the density contrast at the seabed on short-wavelength gravity, ship soundings are used as well, as bathymetry is not knowable over long wavelengths from gravity and the bathymetry-gravity relationship depends on the seabed density used. An explanation for this anomaly may be an erroneous ship track, but the authors can check the locations of satellite fixes that they used from the NGDC file. This kind of anomaly is quite common, so I'm not sure it is worth putting so much emphasis on it.

There is some confusion between the terms magmatic and volcanic. These terms cannot be used interchangeably. For example:

Line 302: authors refer to magmatic, should be volcanic

Line 326: magmatic should be replaced with volcanic

Line 357: magmatic should be replaced with volcanic

Line 323: according to which echosounder data? the multibeam and sub bottom profilers are both echo sounders.

Line 380: High backscatter is not evidence for recent submarine volcanic activity. At best it is evidence for hard seafloor, which may be volcanic. Without samples I don't believe it can be interpreted this way. See comments on backscatter.

Line 416: "we cannot prove or disprove" - I feel that using the word "prove" is a bit dangerous. You have evidence which supports or does not support mantle plume – ridge interaction. I don't think you can prove this.

Figure 7 – This is still too small to see. I suggest zooming right in to the feature described in the text and removing the rest of the line. Alternatively, like is done in 7b, you could have a zoomed in section for each figure. There are a lot of features that are described that are not easy to see in the current format. It is not necessary to show the entire line if you don't need to. Also please label the features you refer to in text to make it easier for readers to keep up.

Figure 8 - it is not clear to me what has been modified or updated on this figure based on the results presented in this study. The readers want to know what has been gained from the current study and applied to this figure. If nothing is added with the dataset presented in this study, I'm not sure this necessary, it is rather a repeat of previous work.

Line 469 – 471 – This is the first explanation of what the BTM is and how its used. It probably needs to come earlier in the manuscript (even though it is detailed in the supplementary material).

We are very grateful for the numerous comments and constructive criticism of reviewer #4.

Reviewer #4 (Remarks to the Author):

As mentioned in previous reviews, the manuscript that is now titled “Young Pre-Shield Volcanism above the Tristan da Cunha Mantle Plume” presents a suite of recently collected and high-quality geophysical data across the westernmost region of the Walvis Ridge west and southwest of Tristan de Cunha. These geophysical data nicely show numerous seafloor features including abyssal hill fabric, seafloor volcanism and core complexes.

After reviewing the revised manuscript and reading the response to reviewers I still feel that this manuscript requires some additional revisions. I have summarised 4 main issues with this revised manuscript, providing examples of each issue. These issues include: 1) the literature as recounted in this manuscript, 2) the use and interpretation of acoustic seafloor backscatter, 3) speculations made in the discussion, and 4) writing style. I have then detailed more minor issues with the manuscript below.

Major issues:

1. Literature reviewed

In Line 61 – 65, the authors present a very brief literature review of the volcanism has a few substantial errors. Basanite does not equal trachybasalt these are different rock types. This is what is implied with “volcanic rocks on the island are dominated by basanites (trachybasalts)...”.

→ Action: Baker et al. used the term “trachybasalt” for the rocks now classified as basanites. We clarified that point.

The summary of the volcanology according to Hicks et al. 2012 is geologically inaccurate. I recommend going back and thoroughly reviewing the original works and that have described the volcanology of this island (which the authors have cited but incorrectly summarised).

The interpretation made by the authors that this is a shield volcano is incorrect, this is not a shield volcano. Hicks et al. 2012 do not describe this a shield volcano, this is a misinterpretation of the literature. This is an oceanic stratovolcano. This literature summary of the volcanism is an extremely important section considering the title of the paper and so should be done thoroughly and so it is geologically correct. I suggest this section is deleted and completely rewritten. Perhaps a greater emphasis on the geology literature surrounding TdC would help. It is troubling to me that this has not been clearly achieved within the introduction. Such an erroneous review of the literature so early on does not set a high bar for the rest manuscript.

→ Action: We thank Reviewer #4 for pointing out that we wrongly used the term “shield volcano”. This mistake was introduced at an early stage of manuscript drafting, and escaped our later proof-reading. We revised but did not completely rewrite this section.

2. Backscatter presentation and interpretation

I have a general issue with the way acoustic seafloor backscatter data is presented and discussed in this manuscript:

2.1 Presentation of backscatter

As I’ve mentioned before, the colour scale is non-standard. I suggested this to be changed in the previous review and it wasn’t “for purposes of classification”. I don’t know where these classifications are, do you mean the figure that has a graduated colour scale? I reiterate that

backscatter should be presented in standard colours so people who use it frequently are familiar.

→ Action: Reviewer #4 may have overlooked that we previously provided a supplementary figure using grey scale and a “white to brown” color ramp, as is often used for sidescan mosaics. Beyond this, we are not aware of any official conventions or standards in the display of backscatter data. Our updated materials also relate clearly how the color ramp we use is for categorised dB-ranges within natural breaks to emphasize rather small areas with high/higher intensity in contrast to larger areas with lower intensity.

We are by no means alone in this kind of approach. To illustrate this, we would like to draw attention to Eason et al (2016) where backscatter is shown in grey scale as well as color scale (blue to green).

We now describe the above mentioned within a separate supplementary method chapter focusing to backscatter intensity respectively high-resolution seabed image reflectivity

When referring to backscatter is it clearer to say “high/low acoustic seafloor backscatter intensity” rather than high/low amplitude backscatter. It’s important to emphasise that this is seafloor backscatter too, as there is more than one type of backscatter data when using EM122 data (e.g. water column backscatter). For clarity and consistency with other literature, I think it’s important to change the way you refer to backscatter.

→ Action: We agree with the need to clarify that we only deal with seafloor backscatter intensity and have altered our text accordingly.

The authors have used exact decibel values in this manuscript, whilst I understand you can compare decibel values within one voyage, it is not valid to compare absolute backscatter values between voyages. Seeing as it appears from the methods section that multiple MBES were used on multiple voyages, and there is no information about backscatter calibration then comparing dB values is not valid in this instance. Using absolute dB values is a bit misleading in any case. I believe it is more accurate to describe backscatter as relatively high/low intensity, I suggest removing any references to absolute acoustic backscatter values in decibels.

→ Action: We have prepared a new separate supplementary method chapter focusing on our use of backscatter intensity data and high-resolution seabed image reflectivity. This includes a statistical treatment of the differences between dB responses in the various systems used and our correction for them.

2.2 Interpretation of backscatter

The way that backscatter is overinterpreted in this manuscript is very concerning to me. For example:

Line 116-118: “High amplitude backscatter, as shown in Fig. 3b, indicates the presence of coarse material, possibly volcanic sediments”

→ Action: We have reconsidered our interpretations of backscatter and, wherever we remain confident in them, we now provide additional and supporting information either in the manuscript or supplementary material.

Line 119-121: “Low backscatter amplitudes indicate that the aprons consist of generally fine-

grained volcanoclastic intermixed or draped with pelagic or hemipelagic sediments”
There are no samples presented in this study so the readers can only assume you are interpreting geology based purely on backscatter intensity. This is highly problematic. There are examples where backscatter has been correlated with seafloor substrates, however ONLY where adequate ground truth data is available. Seafloor backscatter without ground truth and without proper calibration and processing, provides extra information about the hardness of the seafloor and can be used to INFER substrate. High backscatter does not indicate the presence of coarse material, similarly, low backscatter does not always indicate fine-grained material and does not allow you to infer sediment composition. From the descriptions of the seafloor geology it seems like there are geologic samples, but they are not presented?

→ Action: Whilst there are no geological samples available from the region, we make clear in the revised text how our backscatter interpretations are consistent with interpretations made in sub-bottom profiler data and with ground-truthed studies from comparable regions further north in the Atlantic.

I suggest the authors re-evaluate of the way that backscatter is presented to be more consistent with other papers and most importantly revise the interpretation of the backscatter data to reflect what these data can accurately tell us. Alternatively present the geology that backs up your geophysical data.

3. Speculation and discussion

There is a lot of speculation and the discussion does not feed back into the results presented. As a reader, when I get to the discussion, I want to know what the data presented tells us (that we didn't know before) about the volcanism in the region. I don't feel this has been achieved.

→ Action: We have worked on the discussion section to emphasise the new knowledge our data provide about the occurrence of quite young volcanism west of Tristan da Cunha.

The data as they are presented in this study, can tell us about the surface and shallow subsurface geomorphology of the region. Not the age and not the rock/sediment composition. I think that it is important not to overstate what the data is telling us.

→ Action: We disagree with reviewer #4 in this point. Using a wide range of data sets, we can clearly make the point that we have very likely mapped areas that were overprinted by fairly young volcanism (volcanic cones and lava flows).

Discussing the ages of volcanism as being “young” because they “look young” is fraught with complications. It is a dangerous speculation since time and time again Argon dating is showing that all age progressions indicated by edifice shape/preservation are not in fact real or not as linear as they were speculated to be. A hard seafloor alone, without any supporting evidence for age does not automatically equal young volcanism in my opinion.

→ Action: Here, too, we respectfully disagree with reviewer #4. Even if we cannot give exact ages, we can give relative ones, as we document with reference to the sub-bottom profiler data. This, in addition to recent findings of ongoing volcanism on the basis of other geophysical data, allows us to have considerable confidence in the interpretation that fairly young volcanism has indeed taken place over anomalous upper mantle west of Tristan da Cunha.

4. Writing style

The writing style needs work. It is quite informal, repetitive and in places very hard to follow due to the long sentences.

For example: Line 199 – 202: “Evidence for volcanic activity in the form of circular feature and W-E oriented ridges seems to extend out to 250 km westwards of the TdC island group to about 14 48’W, where we mapped a small volcanic cone on top of old abyssal hills (~10 Myr old ocean crust).”

→ Action: As requested in the first round of revisions, a native speaker commented extensively on our manuscript, and has done so a second time in the light of this review comment.

Also, there are a few sentences that don’t seem to add much value – the authors should be more concise with writing, particularly for a high impact journal such as Nature. If it’s not relevant, perhaps leave it out. e.g., line 99-101:” Further away from the islands and the MAR, bathymetry data were collected along the cruise tracks in-between seismological and electromagnetic ocean-bottom stations and on transit to the area36,37” – to me this sentence adds nothing.

→ No action: It was important for us to highlight the limited time available, which was the reason for our non-systematic surveying. Our aim was not only to present interesting results, but also to document the way the data and results were achieved.

Line 412 “although this region is less well covered by bathymetry data” – the wording here could be revised to be clearer.

→ Action: revised.

Repetitive use of “the high-resolution data set” to begin sentences could be changed to make it easier to read.

→ Action: revised.

Further to this, the use of acronyms in this manuscript results in a highly convoluted confusing read. e.g., Line 417 – “The absence of an OCC at the inside corner of the northern RTI on the TdCTFFZS suggests melt supply there may be slightly enhanced by the presence of the Tristan mantle plume.” Acronyms are useful when trying to avoid long confusing sentences with repetitive concepts. The use of acronyms here isolates and confuses the reader, making this much harder work to read. It is in the best interests of the authors to write a clear and concise manuscript, so people want to read (and cite) it.

→ Action: We agree in this point with reviewer #4, and have reduced the number and complexity of abbreviations.

Minor issues:

The title capitalises Pre-Shield, Volcanism, Mantle and Plume. This isn’t necessary. The title should be “Young pre-shield volcanism above the Tristan da Cunha mantle plume”

→ Action: done.

Line 20: change potential to potentially

→ Action: done.

Line 22: “with multibeam and sediment echosoundings” – I think it is more correct (and more common) to say multibeam echosounders (MBES) and sub-bottom profiler. Its good to call these tools the same as what other people call them so everyone is on the same page.

→ Action: done.

Line 38: “The islands are the emergent parts of well-developed shield, post-shield, or even rejuvenated volcanic edifices¹. However, little is known about embryonic stages of ocean-island growth, which begin with the very first eruptions of hot spot-related volcanic rocks onto “pristine” oceanic crust.” – this doesn’t make sense to me, “However” is a conjunction that should be used to convey a change from the last topic. to make this “however” necessary you need to say something like “The islands are the emergent parts of well-developed shield, post-shield, or even rejuvenated volcanic edifices that occur in the latter stages of ocean island growth. However, little is known about embryonic stages of ocean-island growth, which begin with the very first eruptions of hot spot-related volcanic rocks onto “pristine” oceanic crust.”

Or get rid of the however.

→ Action: done.

Line 49: change to ” ...melting relate to the fundamentals of mantle plume dynamics”

→ Action: done.

Line 118: what rise are the authors referring to here?

→ Action: we have clarified which feature it is we refer to.

Line 127-128: “The lobe within the canyon between the TdC and Inaccessible Island is probably the most recently formed” – it is not clear to me what this interpretation is based on. If its only backscatter, then I think it needs to be rethought.

→ Action: during the revision, most of this topic was transferred to the supplementary material since it only complements the main topics of the manuscript.

Line 133: 12 x 20 km =about200 km²? Where is the 200 km coming from? 25 x 40 is closer to 1000 km². Am I missing something here?

→ Action: The mistake, which was carried over from an early version of the manuscript, has been corrected

Features in the text must be clearly labelled on the figures. There are several locations where the authors are referring to a feature which is not labelled on any figure. These should be

labelled to make it easier to follow, some examples that are not labelled include:

Line 161 – the SE flank that has been removed

→ Action: The feature is now labelled more prominently.

Line 164 – the volcanic ligament and complex

→ Action: The text was revised.

Line 234 – spreading compartment corridor

→ No action: The text states clearly where the feature is to be found using geographic coordinates.

Line 254 – normal abyssal hill fabric

→ No action: It is clear from the context of this sentence where this abyssal hill fabric can be observed in the vicinity of the labelled OCC.

Line 316 – faults observed in the sub bottom profiler data

→ Action: revised.

Also important to specifically refer to figures for example Figs 1a and 2b not just Figs 1 & 2

→ Action: revised.

The volcanic structures described in 177-180: have been interpreted to follow a specific trend, which is fine. I feel as though this could be easily backed up with more robust statistical information rather than just having a qualitative comment about how they “seem” to trend. Again, in line 405: “it seems that the strike of the abyssal hills...”. This isn’t very scientific.

→ Action: Checked, rewritten and attributed with more precise information.

Line 218: change 3.0 km to 3 km, and the 1.0 – 2.0 km to 1-2 km. the decimal place adds nothing,

→ Action: done.

Line 261: I think the authors need to be more specific about the source of the earthquake epicentres. At the very least spell out the acronym NEIC and provide a website or location of the data.

→ Action: revised.

The differences in satellite derived bathymetry and multibeam bathymetry described in Lines 272 – 277 are problematic. Bathymetry is predicted with the aid of both satellite altimetry measurements and soundings along ship tracks (e.g., Smith & Sandwell). Although the gravity field is used to predict bathymetry, using the dominating effect of the density contrast at the seabed on short-wavelength gravity, ship soundings are used as well, as bathymetry is

not knowable over long wavelengths from gravity and the bathymetry-gravity relationship depends on the seabed density used. An explanation for this anomaly may be an erroneous ship track, but the authors can check the locations of satellite fixes that they used from the NGDC file. This kind of anomaly is quite common, so I'm not sure it is worth putting so much emphasis on it.

→ Action: We agree with this point and its content, and have added to the supplementary material to take some account of it.

There is some confusion between the terms magmatic and volcanic. These terms cannot be used interchangeably. For example:

Line 302: authors refer to magmatic, should be volcanic

→ Action: done.

Line 326: magmatic should be replaced with volcanic

→ Action: done.

Line 357: magmatic should be replaced with volcanic

→ Action: changed to tectono-magmatic. It should not be “volcanic”.

Line 323: according to which echosounder data? the multibeam and sub bottom profilers are both echo sounders.

→ Action: revised.

Line 380: High backscatter is not evidence for recent submarine volcanic activity. At best it is evidence for hard seafloor, which may be volcanic. Without samples I don't believe it can be interpreted this way. See comments on backscatter.

→ Action: We disagree with reviewer #4 based on our own data, since it is not the backscatter alone that leads us to our interpretation. To make this clearer in the revised text, we describe ancillary and further information (for example, sub-bottom profiler data) that supports our interpretation and provide references and figures for it.

Line 416: “we cannot prove or disprove” - I feel that using the word “prove” is a bit dangerous. You have evidence which supports or does not support mantle plume – ridge interaction. I don't think you can prove this.

→ Action: revised.

Figure 7 – This is still too small to see. I suggest zooming right in to the feature described in the text and removing the rest of the line. Alternatively, like is done in 7b, you could have a zoomed in section for each figure. There are a lot of features that are described that are not easy to see in the current format. It is not necessary to show the entire line if you don't need to. Also please label the features you refer to in text to make it easier for readers to keep up.

→ Action: We want to retain our portrayal of the profiles at large scale to represent the various types of seafloor in the area. To account for the reviewer's comment, which is helpful and constructive, we added more zoom-ins to illustrate specific features.

Figure 8 - it is not clear to me what has been modified or updated on this figure based on the results presented in this study. The readers want to know what has been gained from the current study and applied to this figure. If nothing is added with the dataset presented in this study, I'm not sure this necessary, it is rather a repeat of previous work.

→ Action: revised. The new features are the signs of fairly recent volcanism on top of the known structures and the existence of an oceanic core complex.

Line 469 – 471 – This is the first explanation of what the BTM is and how its used. It probably needs to come earlier in the manuscript (even though it is detailed in the supplementary material).

→ Action: revised. We now make clear that the BTM is a GIS tool already in the introduction.

REVIEWERS' COMMENTS:

Reviewer #5 (Remarks to the Author):

Review by William Chadwick

General comments on the manuscript:

The editor asked me to review the revisions made by the authors in response to the latest comments by Reviewer #4, who is unfortunately no longer available to evaluate them. To do this, I have read the manuscript and added comments into the "rebuttal" PDF that the journal provided with the itemized comments from Reviewer #4 and the responses from the authors. I also did my own review of the revised manuscript and have made some suggested edits and comments in an annotated version of the manuscript. In general, the latest revisions adequately respond to most of Reviewer #4's comments, but there are still a few issues that need addressing before publication. The most important of these are the "interpretation of backscatter" (2.2 in the rebuttal PDF), and "speculation and discussion" as it relates to the age of seafloor deposits (3.0 in the rebuttal PDF). Both of these issues can be addressed relatively easily by just rephrasing with appropriate qualifying words to clarify that there is fundamental uncertainty and ambiguity when interpreting backscatter data without ground truth from seafloor samples. While this is relatively easy to resolve, it is also critically important in order to avoid the overinterpretation of data that is not supported by the scientific evidence. It is so important that the paper should not be published without these further (minor) revisions to address these concerns. There are also some sentences that seem either unnecessary or overly speculative that I think should be omitted to make the paper clearer and more concise. Therefore, I recommend publication after another round of minor revisions.

Selected specific comments linked to line numbers in the annotated manuscript:

Lines 59 & 70: Labels in Figures 2 & 3 could be in larger font to make them easier to read.

Line 86: Here and elsewhere, the age of seafloor features or deposits must be stated in relative terms, such as "relatively recent" or "relatively young" instead of "recent" or "young", which imply some knowledge of absolute age and can be easily misinterpreted. The data presented in this paper provide good evidence for relative age, but only loose constraints on absolute age.

Line 116: I would encourage the authors to include grey-scale versions of Figures 2 AND 3, 4, and, 6 in the Supplementary figures. The value of a grey scale presentation of backscatter data is that it presents the data in an unbiased and uninterpreted manner, which allows the reader to better evaluate the interpretations of the authors.

Line 117: Similarly, here and elsewhere, the interpretation of backscatter data needs to acknowledge more uncertainty. For example, "... backscatter intensities ... indicate a coarse, rough, and/or hard seafloor ..." should be changed to "... backscatter intensities ... suggest a relatively coarse, rough, and/or hard seafloor ..."

Line 123: I don't find the figure of rugosity (Fig. 3c) very informative and I think it could be omitted. Also, the existence and interpretation of the slope failure features is not super obvious to me, at least looking at the resolution of the figures available for review.

Lines 141-142: This ridge is not obvious in the figures (at least in the resolution I have). Omit?

Lines 148-149: The crater discussed is not visible in Figure 3.

Lines 163-165: "heaps" is not a common term in volcanology or marine geology. Omit.

Line 167: "Towards the west end of the study area". What area is being referred to here? West part of A & B? Or west of A & B? Or west edge of Figure 3? Or west edge of Figure 2? This doesn't seem to be important. Omit?

Lines 168-169: Again, "ridge mounds" is not a common term in volcanology or marine geology. Omit.

Lines 171-174: This paragraph seems unnecessary. Omit?

Lines 238-243: This seems like unnecessary speculation. Omit.

Lines 274-277: This seems like a stretch. Even though the manufacturer says the sub-bottom data has a resolution of 0.15 m, to me that means that is the effective pixel size of the sub-bottom cross-sections. That is NOT the same thing as saying you can detect a layer of sediment on the seafloor that is only 15 cm thick. I think this is an extreme understatement of the uncertainty in the data (or an overstatement of the resolution). In any case, using age descriptions such as "very young" or "very recent" for deposits that could be 50,000 years old (or more) is unwarranted. Again, I think all discussion of age in this paper should be RELATIVE - like saying "relatively young" or "relatively recent".

Line 348-352: Need to qualify these interpretations more.

Lines 352-359: Omit? This doesn't seem to add much.

Lines 365-369: This is speculative and doesn't add much. This just weakens the paper as a whole and should be omitted.

Lines 388-389: Unnecessary. Omit.

Line 414: The depth of Isolde's summit is not much of a constraint because the 2012 Havre eruption (Kermadec arc) formed a large pumice raft from eruptive vents deeper than 900 m.

Line 415-419: Too much speculation. Omit.

Line 452: Say what "shallow seafloor" means with a depth range.

Lines 481-487: I think you could omit this text about Rugosity. I don't think Figure 3c adds much to the paper that isn't evident in the other figures, so all this could be removed if Fig. 3c is also omitted.

Line 562: Note correction to reference 4 title.

Line 820: For suggested edits to the figure captions, see the caption text below each figure (for example, line 886, etc)

We are very grateful for the numerous comments and constructive criticism of reviewer #5. We integrated most of his comments and suggestions (also from the annotated manuscript).

Reviewer #5 (Remarks to the Author):

Review by William Chadwick

General comments on the manuscript:

The editor asked me to review the revisions made by the authors in response to the latest comments by Reviewer #4, who is unfortunately no longer available to evaluate them. To do this, I have read the manuscript and added comments into the “rebuttal” PDF that the journal provided with the itemized comments from Reviewer #4 and the responses from the authors. I also did my own review of the revised manuscript and have made some suggested edits and comments in an annotated version of the manuscript.

In general, the latest revisions adequately respond to most of Reviewer #4’s comments, but there are still a few issues that need addressing before publication. The most important of these are the “interpretation of backscatter” (2.2 in the rebuttal PDF), and “speculation and discussion” as it relates to the age of seafloor deposits (3.0 in the rebuttal PDF). Both of these issues can be addressed relatively easily by just rephrasing with appropriate qualifying words to clarify that there is fundamental uncertainty and ambiguity when interpreting backscatter data without ground truth from seafloor samples. While this is relatively easy to resolve, it is also critically important in order to avoid the overinterpretation of data that is not supported by the scientific evidence. It is so important that the paper should not be published without these further (minor) revisions to address these concerns. There are also some sentences that seem either unnecessary or overly speculative that I think should be omitted to make the paper clearer and more concise.

Therefore, I recommend publication after another round of minor revisions.

→ Action: As outlined in response to more specific comment below, we have adopted the use of terms like ‘relatively young’ and ‘relatively recent’ to qualify our interpretations and make clear that we have no absolute dating material.

Selected specific comments linked to line numbers in the annotated manuscript:

Lines 59 & 70: Labels in Figures 2 & 3 could be in larger font to make them easier to read.

→ Action: We increased the labels where possible and expect them to be legible now.

Line 86: Here and elsewhere, the age of seafloor features or deposits must be stated in relative terms, such as “relatively recent” or “relatively young” instead of “recent” or “young”, which imply some knowledge of absolute age and can be easily misinterpreted. The data presented in this paper provide good evidence for relative age, but only loose constraints on absolute age.

→ Action (see also main comment above): We now use the terms as suggested by reviewer #5.

Line 116: I would encourage the authors to include grey-scale versions of Figures 2 AND 3, 4, and, 6 in the Supplementary figures. The value of a grey scale presentation of backscatter data is that it presents the data in an unbiased and uninterpreted manner, which allows the reader to better evaluate the interpretations of the authors.

→ Action: We now include grey-scale backscatter images for all regions within the supplementary material, as well as providing the original mosaic for download via the PANGAEA portal.

Line 117: Similarly, here and elsewhere, the interpretation of backscatter data needs to acknowledge more uncertainty. For example, "... backscatter intensities ... indicate a coarse, rough, and/or hard seafloor ..." should be changed to "... backscatter intensities ... suggest a relatively coarse, rough, and/or hard seafloor ..."

→ Action: We followed the suggestions of reviewer #5 and changed the wording accordingly.

Line 123: I don't find the figure of rugosity (Fig. 3c) very informative and I think it could be omitted. Also, the existence and interpretation of the slope failure features is not super obvious to me, at least looking at the resolution of the figures available for review.

→ No action: We intend to keep the figure showing rugosity because of its importance for building confidence in our interpretations. Rugosity is an important derivative to assure quality control because it complements, and permits comparison to, other products such as BTM or slope. As with the fine and broad scale BPI, rugosity can highlight local and regional or even broad-scale areas of different kind of roughness, based on different bathymetry grid cell sizes.

We also retain the interpretation of slope failure features, which reinforces previous alternative interpretations. Although slope failure is not the main focus of our manuscript, it is undoubtedly an important process for volcanic evolution, and thus certainly of relevance to the manuscript's overall theme. In view of its ancillary importance and the reviewer's skepticism about the evidence, we restrict this interpretation to the supplementary material

Lines 141-142: This ridge is not obvious in the figures (at least in the resolution I have).
Omit?

→ Action: The feature was already well visible to us at the resolution we provided Figure 3(d) (now Fig. 5) at. To improve our illustration, we have added new hillshade-illuminated zoom-in panels for three of the landmarks described in the results section.

Lines 148-149: The crater discussed is not visible in Figure 3.

→ Action: We include a new zoom-in of the summit region of Isolde Seamount to better illustrate this interpretation.

Lines 163-165: "heaps" is not a common term in volcanology or marine geology. Omit.

→ Action: We have removed this term.

Line 167: "Towards the west end of the study area". What area is being referred to here? West part of A & B? Or west of A & B? Or west edge of Figure 3? Or west edge of Figure 2? This doesn't seem to be important. Omit?

→ Action: We have revised the wording so it is easier to understand what is being referred to.

Lines 168-169: Again, "ridge mounds" is not a common term in volcanology or marine geology. Omit.

→ Action: We have removed this term.

Lines 171-174: This paragraph seems unnecessary. Omit?

→ Action: We have removed most of the paragraph except the first sentence, which serves as a general statement.

Lines 238-243: This seems like unnecessary speculation. Omit.

→ Action: We have removed the speculation.

Lines 274-277: This seems like a stretch. Even though the manufacturer says the sub-bottom data has a resolution of 0.15 m, to me that means that is the effective pixel size of the sub-bottom cross-sections. That is NOT the same thing as saying you can detect a layer of sediment on the seafloor that is only 15 cm thick. I think this is an extreme understatement of the uncertainty in the data (or an overstatement of the resolution).

→ Action: To counter the impression that the resolution figure we used 'seems like a stretch', we have added a detailed description of the calculations used to estimate vertical resolution in sub-bottom profiler data in the Methods section.

In any case, using age descriptions such as "very young" or "very recent" for deposits that could be 50,000 years old (or more) is unwarranted. Again, I think all discussion of age in this paper should be RELATIVE - like saying "relatively young" or "relatively recent".

→ Action: As outlined above, we now use the ‘relative’ terms as suggested by reviewer #5.

Line 348-352: Need to qualify these interpretations more.

→ Action: We made minor changes here and gave references to published data that provide the basis of our interpretation in order to avoid repeating interpretations stated in these references.

“Slower average S-wave seismic velocities in the upper mantle were also observed in this area by Bonadio et al.³³ by analyzing teleseismic surface waves. Among the range of possible causes for relatively recent tectono-magmatic activity between TdC and the MAR, these seismic anomalies independently support the presence of warm and melt-rich (<1%³³) upper mantle.”

Lines 352-359: Omit? This doesn't seem to add much.

→ No Action: We kept these sentences because they are necessary to describe the data/references behind the schematic sketch.

Lines 365-369: This is speculative and doesn't add much. This just weakens the paper as a whole and should be omitted.

→ Action: We kept the observation (now in the Results section) but omitted the speculative interpretation.

Lines 388-389: Unnecessary. Omit.

→ Action: We rewrote this segment in order to better express its necessity.

“They are, in all cases, products of cooler spreading regimes with lower melt supplies at the ridge axis, suggesting the absence or intermittency of plume influence on the MAR south of the TTFZ.”

Line 414: The depth of Isolde’s summit is not much of a constraint because the 2012 Havre eruption (Kermadec arc) formed a large pumice raft from eruptive vents deeper than 900 m.

→ Action: We revised the paragraph to emphasize the summit depth is well within the range of depths from which pumice can be delivered.

Line 415-419: Too much speculation. Omit.

→ Action: We consider this as our interpretation of the recorded bathymetry and seismological data rather than speculation. Therefore, we keep this sentence.

Line 452: Say what "shallow seafloor" means with a depth range.

→ Action: We specified "shallow seafloor" as seafloor with depths <200 mbsl.

Lines 481-487: I think you could omit this text about Rugosity. I don't think Figure 3c adds much to the paper that isn't evident in the other figures, so all this could be removed if Fig. 3c is also omitted.

→ Action. See the reply to the comment on line 123, given above.

Line 562: Note correction to reference 4 title.

→ Action: The title of the reference was corrected.

Line 820: For suggested edits to the figure captions, see the caption text below each figure (for example, line 886, etc)

→ Action: The suggested edits were introduced in the revised version.